# High clonal diversity and spatial genetic admixture in early prostate cancer and surrounding normal tissue

Ning Zhang[1,2,14,15], Luuk Harbers [1,2,15], Michele Simonetti [1,2,15], Constantin Diekmann [1,2], Quentin Verron[1,2], Enrico Berrino [3,4], Sara E. Bellomo[3,5], Gabriel M. C. Longo [6], Michael Ratz [7], Niklas Schultz[2,8], Firas Tarish[9], Peng Su[10], Bo Han[10], Wanzhong Wang[8], Sofia Onorato[1,2], Dora Grassini[4], Roberto Ballarino [1,2], Silvia Giordano [3,5], Qifeng Yang[11], Anna Sapino [3,4], Jonas Frisén [7], Kanar Alkass[7,8], Henrik Druid [8], Vassilis Roukos [6,12], Thomas Helleday [2,8], Caterina Marchiò[3,4], Magda Bienko [1,2,13] & Nicola Crosetto [1,2,13] ✉

Somatic copy number alterations (SCNAs) are pervasive in advanced human cancers, but their prevalence and spatial distribution in early-stage, localized tumors and their surrounding normal tissues are poorly characterized. Here, we perform multi-region, single-cell DNA sequencing to characterize the SCNA landscape across tumor-rich and normal tissue in two male patients with localized prostate cancer. We identify two distinct karyotypes: 'pseudo-diploid' cells harboring few SCNAs and highly aneuploid cells. Pseudo-diploid cells form numerous small-sized subclones ranging from highly spatially localized to broadly spread subclones. In contrast, aneuploid cells do not form subclones and are detected throughout the prostate, including normal tissue regions. Highly localized pseudo-diploid subclones are confined within tumor-rich regions and carry deletions in multiple tumor-suppressor genes. Our study reveals that SCNAs are widespread in normal and tumor regions across the prostate in localized prostate cancer patients and suggests that a subset of pseudo-diploid cells drive tumorigenesis in the aging prostate.

Somatic copy number alterations (SCNAs) are pervasive in human cancers[1–3]. It has been proposed that SCNAs form in the early stages of tumorigenesis in punctuated evolutionary bursts[4–7]. Indeed, SCNAs have been detected in pre-malignant lesions such as Barret's esophagus years before the onset of invasive esophageal cancer[8] and oral leukoplakias undergoing progressive transformation to oral squamous cell carcinomas[9]. However, the exact timing and mechanisms of SCNA formation during carcinogenesis remain poorly

[1]Department of Microbiology, Tumor and Cell Biology, Karolinska Institutet, Stockholm 17177, Sweden. [2]Science for Life Laboratory, Stockholm 17177, Sweden. [3]Candiolo Cancer Institute, FPO – IRCCS, Candiolo, SP142, km 3,95, 10060 Turin, Italy. [4]Department of Medical Sciences, University of Turin, Turin, Italy. [5]Department of Oncology, University of Turin, Turin, Italy. [6]Institute of Molecular Biology (IMB), Mainz 55128, Germany. [7]Department of Cell and Molecular Biology, Karolinska Institute, Stockholm 17177, Sweden. [8]Department of Oncology and Pathology, Karolinska Institutet, Stockholm 17177, Sweden. [9]Södersjukhuset, Stockholm 11883, Sweden. [10]Department of Pathology, Qilu Hospital of Shandong University, Ji'nan 250012, China. [11]Department of Breast Surgery, General Surgery, Qilu Hospital of Shandong University, Ji'nan 250012, China. [12]Department of General Biology, Medical School, University of Patras, Patras, Greece. [13]Human Technopole, Viale Rita Levi-Montalcini 1, 20157 Milan, Italy. [14]Present address: Department of Breast Surgery, General Surgery, Qilu Hospital of Shandong University, Ji'nan 250012, China. [15]These authors contributed equally: Ning Zhang, Luuk Harbers, Michele Simonetti. ✉e-mail: nicola.crosetto@ki.se

understood and a map of SCNAs in early-stage, localized tumors and their surrounding normal tissues is missing.

In prostate cancers, SCNAs represent the predominant type of genomic alteration and are thought to be the major driver of these tumors[10]. Early-stage localized prostate cancers are often characterized by the presence of multiple spatially confined tumor foci[11]. Thus, these tumors represent an ideal model to investigate the spatial distribution of SCNAs in early-stage cancer and adjacent normal tissue. Multi-region sequencing of medium-grade (Gleason 7) localized prostate cancers previously showed extensive intratumor heterogeneity between distinct tumor foci at the level of single-nucleotide variants (SNVs), SCNAs, and genomic rearrangements, suggesting an independent origin of different foci[12]. Polyclonal origin was also suggested by another study, in which SNVs were detected in morphologically normal prostate tissue distant from cancer lesions, consistent with so-called cancerization field effects[13]. More recently, some of us leveraged spatial transcriptomics (ST)[14] to assess the spatial distribution of SCNAs in different tumor types, including two Gleason 7 prostate cancer samples[15]. By performing ST on tissue sections from multiple tissue blocks excised from a midsection of a prostatectomy sample, large chromosomal amplifications and deletions were detected both in cancerous areas and tissue regions classified as benign[15]. However, since this approach infers SCNAs from RNA-seq data and classical ST does not reach single-cell resolution, a thorough characterization of the diversity of SCNAs across normal and tumor regions was not achieved in that study[15].

In this work, to overcome these limitations and gain further insights into the spatial distribution of SCNAs and mutations in the prostate gland, we perform multi-region single-cell copy number profiling combined with targeted deep sequencing of genomic DNA extracted from the same regions in two prostatectomy samples from patients diagnosed with localized prostate cancer. We show that SCNAs and, to a lesser extent, mutations are widespread throughout the prostate gland, including tissue regions classified as normal based on both morphological assessment and RNA-seq. However, a subpopulation of cells carrying subclonal deletions affecting specific tumor suppressor genes is highly enriched in tumor areas, implicating these subclones in early tumorigenesis. We provide a prostate-wide, spatially resolved, single-cell SCNA map, demonstrating that genome instability is widespread throughout the aging prostate and suggesting that a specific set of high-risk SCNAs paves the path towards tumorigenesis.

## Results

### Study design and methodology for single-cell SCNA profiling
We prospectively collected prostatectomy samples from six patients (P1-P6) diagnosed with localized prostate cancer (age range: 43–65 yrs) and sliced a transversal prostate midsection into multiple small tissue blocks (~125 mm³, hereafter named 'regions') using the same approach that some of us previously adopted for multi-region spatial transcriptomics[15] (Supplementary Table 1 and Methods). We sliced each region into halves and used one half for genomic DNA (gDNA)/RNA extraction and the other for single nuclei isolation. For two samples (P2 and P5) in which the quality of gDNA was high in all the regions (n = 34 and 28 regions, respectively), we used the single nuclei suspensions to profile SCNAs in individual cells, while we used the extracted gDNA for targeted mutation sequencing (Fig. 1a and Methods). In addition, we performed histopathological assessment using two tissue sections sliced, respectively, from the top and bottom of each region and stained with hematoxylin and eosin (Methods). Since in this study prostatectomy was performed endoscopically, the quality of the RNA extracted from each region was not suitable for spatial transcriptomics but sufficiently good for bulk RNA-seq. Therefore, to complement the histopathological classification, we also performed bulk RNA-seq for all the regions of P2 and P5 (Supplementary Methods).

To perform single-cell SCNA profiling from all the regions in a cost-effective manner, we leveraged our CUTseq method for highly multiplexed SCNA profiling[16], and developed a single-cell adaptation of it, which we named single-cell CUTseq or scCUTseq (Methods). A step-by-step scCUTseq protocol is available at Protocol Exchange. Briefly, in scCUTseq, cells or nuclei are sorted into 384-well plates pre-filled with mineral oil to prevent evaporation, and whole genome amplification (WGA) by multiple annealing and looping based amplification cycles (MALBAC)[17] is performed in nanoliter volumes using a contactless nanodispensing device (I.DOT) to minimize reagent volumes and hence costs (Fig. 1b). For sorting, we optimized gating parameters aiming at sorting only single cells or nuclei into each well (Supplementary Fig. 1a–d). Next, each nucleus is barcoded using the same approach as in CUTseq[16] but reagent volumes are downscaled with I.DOT to increase the efficiency of enzymatic reactions and further reduce costs. Lastly, the barcoded nuclei are pooled together, followed by gDNA purification and fragmentation, in vitro transcription to selectively amplify gDNA downstream of CUTseq adapters, library preparation, and sequencing on Illumina platforms (Fig. 1b). A key feature of scCUTseq is its compatibility with fixed cells and nuclei, allowing to immediately stabilize gDNA to prevent batch effects when multiple samples must be processed in parallel and single-nucleus sorting cannot be immediately performed after nuclei extraction, as in this study.

To assess the analytical performance and validate scCUTseq, we performed several experiments. First, we assessed the effect of cell fixation and WGA volume reduction and found that the genome coverage and copy number profiles remained stable down to 1:200 MALBAC reagent volume scaling, independently of cell fixation (Supplementary Fig. 2a–f and Supplementary Methods). Importantly, scCUTseq could clearly resolve the copy number profiles of different cell lines and was free of cross-contamination (Supplementary Fig. 2g, h), highlighting its specificity. Second, we assessed the sensitivity of scCUTseq by determining its ability to detect a 7 megabase (Mb) deletion introduced by CRISPR-Cas9 in a small percentage (~3%) of cells transfected with two small guide RNAs (sgRNA) targeting the KMT2A and HYLS1 loci on chromosome (chr) 11 (Supplementary Fig. 3a and Supplementary Methods). In 3.3% of the cells transfected with both sgRNAs, scCUTseq detected a single copy of the exact 7 Mb deletion, whereas in 3.2% of the cells the deletion extended until the end of the q-arm of chr11, in agreement with similar estimates obtained by DNA fluorescence in situ hybridization (FISH) (Supplementary Fig. 3b–h and Supplementary Methods), thus highlighting the sensitivity of our method. Third, to rule out that the MALBAC step introduces SCNA artefacts, we compared scCUTseq with Acoustic Cell Tagmentation (ACT)[18], another single-cell SCNA profiling method that is based on DNA tagmentation and does not involve a WGA step (Methods). Although ACT performed better in terms of breadth of coverage (as expected, given that scCUTseq utilizes restriction enzymes, while ACT relies on tagmentation) and read count over-dispersion, both methods yielded similar copy number profiles (Supplementary Fig. 4a–h), indicating that the WGA step in scCUTseq does not result in obvious artefactual SCNA calls. Altogether, these results made us confident to apply scCUTseq to systematically profile SCNAs across all the tissue regions of the P2 and P5 prostate samples described above.

### Two major types of SCNA-containing cells populate the prostate of patients with localized prostate cancer
We first classified each region in P2 and P5 as tumor-rich regions (TRRs, >50% of tumor cells), focally enriched regions (FERs, 10–50% of tumor cells), and normal regions (NORs) based on histopathological examination by two board-certified pathologists who assessed the samples independently (Supplementary Figs. 5 and 6). In P2, TRRs and FERs were mainly distributed along the left-posterior prostate margin,

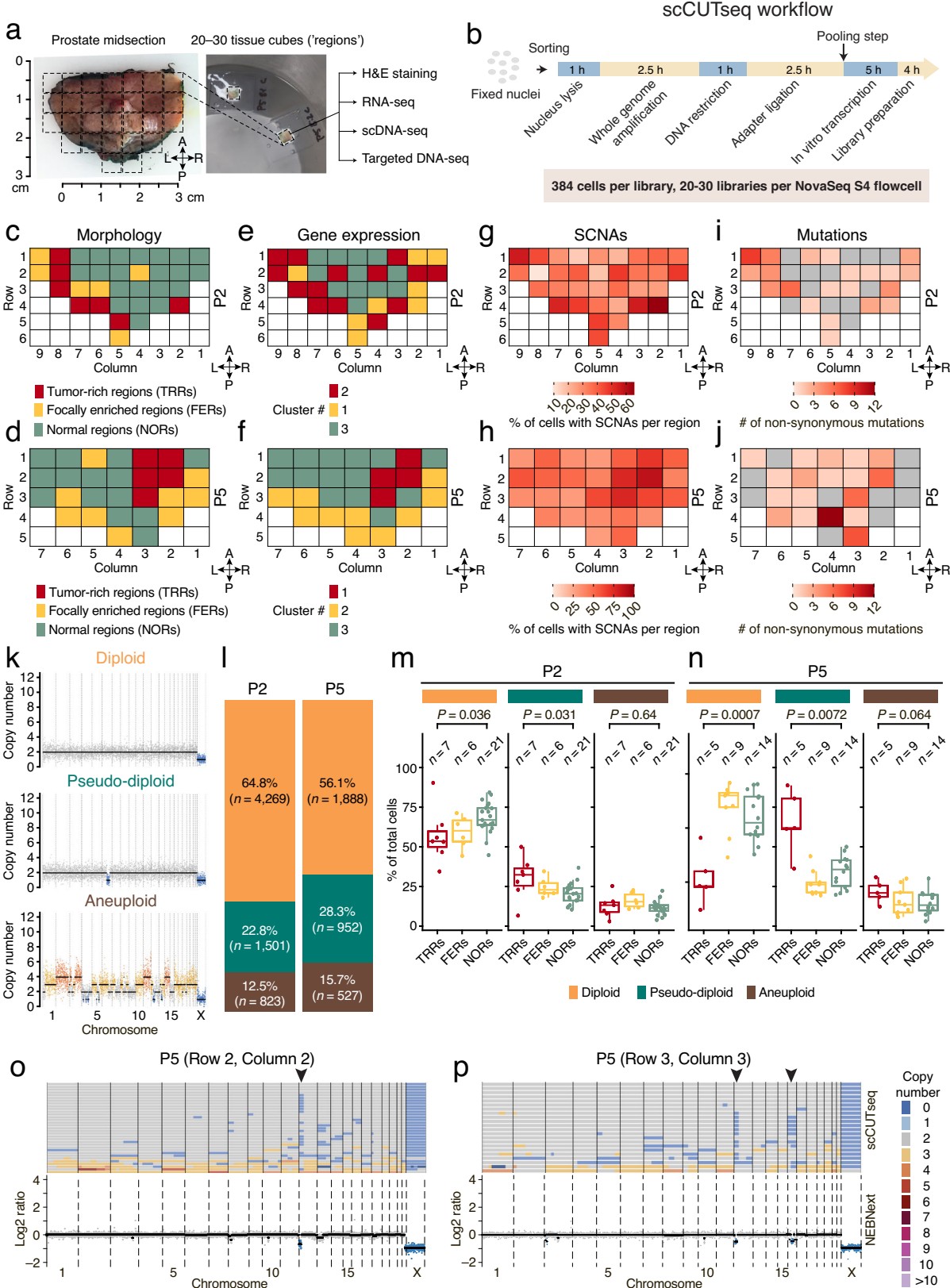

whereas in P5 they formed a clearly visible cluster in the right-anterior part of the prostate (Fig. 1c, d). Clustering of bulk RNA-Seq data using a non-negative matrix factorization approach yielded three distinct clusters that recapitulated, although not completely, the distribution of TRRs, FERs, and NORs (Fig. 1e, f, Supplementary Fig. 7a, b, and Supplementary Methods). Gene set enrichment analysis (GSEA)

revealed that multiple genes associated with prostate cancer were overexpressed in TRRs compared to other regions, including the well-known prostate cancer biomarker *PCA3* gene and the recently identified diagnostic biomarker *TMTC4*, which is highly specific for prostate cancer[19] (Supplementary Figs. 8 and 9, and Supplementary Methods). Notably, especially in sample P2, several regions classified as NORs

**Fig. 1 | SCNAs and mutations are widespread in prostates from patients with localized prostate cancer. a** Photograph of one of the prostate samples (P5) and of two frozen tissue cubes ('regions') excised from the same sample. The dashed grid marks (roughly) the regions in the corresponding maps in (**d, f, h, j**). **b** scCUTseq workflow. **c, d** Schematic maps displaying the histopathological classification of regions of the two prostate samples (P2 and P5, respectively). Each cell in the grid represents a tissue region, white cells indicate absence of tissue. **e, f** Similar to (**c, d**) but displaying the classification of each tissue region based on clustering of RNA-seq data (see Supplementary Fig. 5). **g, h** Percentage of all cells carrying at least one SCNA in each region of P2 and P5, respectively. **i, j** Schematic map showing the number of non-synonymous SNVs in P2 and P5, respectively. The anatomical orientation of the maps in c-j is shown by the four arrows near each map. A, anterior. P, posterior. L, left. R, right. **k** Representative copy number profiles (500 kb resolution) of diploid, pseudo-diploid and aneuploid cells. **l** Fraction of each karyotype shown in (**k**) in P2 and P5. *n*, number of cells. **m, n** Percentage of each karyotype exemplified in (**k**) in tumor-rich regions (TRRs), focally enriched regions (FERs) and normal regions (NORs) in P2 and P5, respectively. *n*, number of regions. *P*, Wilcoxon test, two-tailed. Boxplots extend from the 25th to the 75th percentile, horizontal bars represent the median, and whiskers extend from $-1.5 \times IQR$ to $+1.5 \times IQR$ from the closest quartile. IQR, inter-quartile range. **o** Copy number profiles (500 kb resolution, each row represents a single cell) from nuclei extracted from one region in sample P5 (top) and corresponding bulk copy number profile (500 kb resolution) from the same region (bottom). The arrowhead pinpoints a subclonal deletion that was detected by both scCUTseq and bulk DNA-seq. **p** As in (**o**) but for a different region in sample P5. A link to the Source Data for this figure is provided in the Data Availability statement.

based on histopathology shared similar gene expression profiles with TRRs or FERs, indicating that these regions might already contain a population of tumor cells undetected by histopathology.

Next, we used scCUTseq to generate DNA copy number profiles from 23,808 nuclei from 62 regions in total. To automatically filter out low-quality profiles, we trained a random forest classifier on 2304 manually annotated single-cell profiles, which yielded a very high classification accuracy (area under the curve, AUC: 0.993) (Supplementary Fig. 10a–d, Supplementary Data 1, and Methods). By applying this classifier to our dataset, we obtained a total of 9960 high-quality profiles, including 194 high-quality profiles per region, on average, for P2 (range: 98–362; percentage of nuclei retained for analysis: 25.5–94.3%) and 120 for P5 (range: 17–212; percentage of nuclei retained for analysis: 4.4–55.2%) (Supplementary Fig. 10e, f). Compared to ACT, scCUTseq typically yielded a higher fraction of cells with high-quality copy number profiles from the same region (Supplementary Fig. 10g). Notably, the percentage of cells retained by our classifier was much higher in the case of other tissues that we profiled by scCUTseq (Supplementary Fig. 10h), indicating that the type of tissue and procedure to obtain it can greatly affect the fraction of cells that can be analyzed.

We then wondered whether some of the cells rejected by the random forest classifier are in the S phase of the cell cycle, explaining their noisy copy number profiles. To test this possibility, we leveraged scAbsolute[20], a computational tool which infers the replication status of single cells from scDNA-seq data. Application of scAbsolute to the copy number profiles of 991 cells from four different scCUTseq libraries from P2 and P5 yielded clearly bimodal distributions of the cycling activity inferred by scAbsolute (Supplementary Fig. 11a–d and Methods), indicating that a sizable fraction of the cells in these samples were in S phase. Notably, the cycling activity was significantly higher for cells rejected by the random forest classifier compared to cells with high-quality copy number profiles (Supplementary Fig. 11e). Indeed, the majority of cells excluded were classified as cycling based on scAbsolute (Supplementary Fig. 11f–i), suggesting that even cells with noisy copy number profiles are not necessarily an artefact of scCUTseq but rather reflect a specific biological state. For simplicity, however, we excluded these cells from all downstream analyzes.

Next, we visualized the spatial distribution across P2 and P5 of cells with high-quality copy number profiles carrying SCNAs. To our surprise, in both samples, SCNA-harboring cells were widespread throughout all the regions examined and SCNAs were clearly enriched inside TRRs and FERs (Fig. 1g, h). As a comparison, scCUTseq on 2,304 nuclei (of which 1,703 (73.9%) yielded high-quality copy number profiles) from fresh frozen human forebrain and skeletal muscle autopsy samples from two donors revealed a very small number of SCNAs, and genome-wide copy number profiles were completely flat in most of the cells analyzed (Supplementary Fig. 12a–d, Supplementary Data 2, and Methods), further proving that SCNAs detected by scCUTseq are not an artefact. Furthermore, compared to SCNAs, point mutations were more localized in TRRs and FERs, even though they were also detected in many regions classified as normal (Fig. 1i, j).

We then sought to investigate the SCNAs detected in P2 and P5 samples in depth. Visual inspection of high-quality copy number profiles revealed the existence of three major groups of cells: (i) diploid cells with completely flat copy number profiles; (ii) pseudo-diploid cells harboring a few sparse alterations typically consisting of (sub-)chromosomal arm deletions affecting one or few chromosomes; and (iii) aneuploid cells characterized by multiple whole chromosome amplifications and deletions, reminiscent of so-called 'hopeful monsters' previously described in colorectal cancer[7] (Fig. 1k). Pseudo-diploid cells accounted for 22.8% and 28.3% of all high-quality cells in P2 and P5, respectively, and were significantly enriched in TRRs compared to NORs in the same sample (Fig. 1l–n). Conversely, diploid cells were significantly more abundant in NORs compared to TRRs, as expected (Fig. 1m, n). Aneuploid cells accounted for a lower fraction of the cells (12.5% and 15.7% in P2 and P5, respectively) and were detected at similar frequencies across TRRs, FERs, and NORs (Fig. 1l–n). Notably, we detected similar proportions of diploid, pseudo-diploid, and aneuploid cells in two regions that we profiled by scCUTseq in each of the four additional prostate samples from four other patients collected in this study (Supplementary Fig. 13a, b). Many of the deletions detected by scCUTseq across multiple pseudo-diploid cells were also captured by shallow whole-genome sequencing (WGS) of gDNA extracted from the same regions, while WGS of matched peripheral blood gDNA yielded flat copy number profiles (Fig. 1o, p, Supplementary Fig. 14a, b, and Supplementary Methods), further highlighting the specificity of scCUTseq. We note, however, that smaller deletions present in only a few cells might not have been detected in this validation experiment because of the very low sequencing depth achieved. These results indicate that SCNAs are ubiquitous in the prostate of patients with localized prostate cancer and that pseudo-diploid and aneuploid cells co-exist in these tissues, most likely reflecting different mechanisms of SCNA formation.

## High clonal and spatial heterogeneity of pseudo-diploid cells

Next, we explored whether the pseudo-diploid cells detected throughout the prostate gland are phylogenetically related and form spatially distinct subclones. To this end, we used MEDICC2[21] to generate phylogenetic trees based on single-cell copy number profiles and performed clustering to identify subclones sharing similar profiles. We identified numerous (*n* = 79 and 52 in P2 and P5, respectively) highly divergent subclones (range: 5–56 and 5–52 cells per clone, respectively) mainly harboring whole-chromosome or chromosome arm-level deletions typically affecting less than 3% of the genome (range: 0–435 and 0–239 Mb in P2 and P5, respectively) (Fig. 2a–f and Methods). As a comparison, application of scCUTseq to 2304 nuclei (of which 734 (31.9%) yielded high-quality SCNA profiles) from two breast cancer samples resulted in a substantially smaller number (7 and 3,

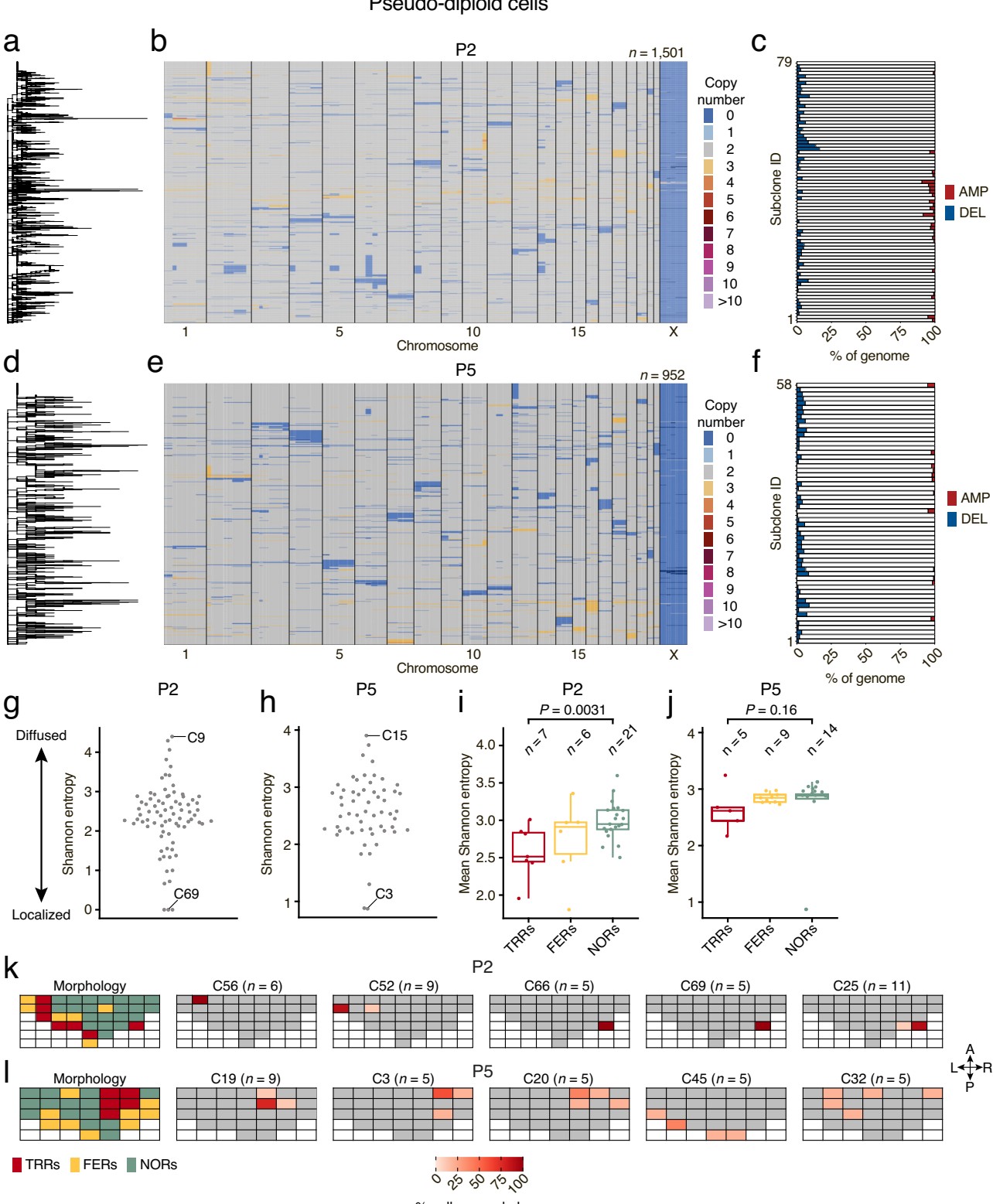

**Pseudo-diploid cells**

respectively) of phylogenetically related subclones identified (Supplementary Fig. 15a–d and Methods). Notably, pseudo-diploid subclones harboring deletions had very few amplification events and vice versa (Fig. 2c, f).

We then assessed how the subclones identified are spatially distributed throughout the regions analyzed. Each region typically featured multiple subclones (range: 5–34 and 1–25 for P2 and P5, respectively) and most of the subclones were detected in more than

one region (Supplementary Fig. 16a, b). Some subclones were highly localized whereas others were spread throughout the entire prostate midsection (Supplementary Figs. 17 and 18). To quantify the observed subclonal spatial heterogeneity, we calculated the Shannon entropy of each subclone across all the regions analyzed, normalizing for the different number of nuclei that were retained by our random forest classifier in each region (Supplementary Fig. 10e, f and Methods). Low Shannon entropy values indicate localization to one or a few regions,

**Fig. 2 | Pseudo-diploid cells are characterized by high clonal and spatial heterogeneity. a** Phylogenetic Newick tree of pseudo-diploid cells identified in sample P2. Each leaf in the tree corresponds to one pseudo-diploid cell. **b** Copy number profiles (500 kb resolution) of the pseudo-diploid cells in the phylogenetic tree shown in (a). *n*, number of cells. **c** Fraction of the genome amplified (AMP) or deleted (DEL) in each subclone identified in P2 and P5, respectively. **d–f** As in (**a–c**) but for prostate sample P5. **g, h** Shannon entropy of each pseudo-diploid cell subclone (C) identified in P2 and P5, respectively. The most localized and widespread subclones in each sample are labeled (see Supplementary Figs. 17 and 18 for the distribution map of each subclone). **i, j** Distribution of the mean Shannon entropy of pseudo-diploid subclones inside tumor-rich regions (TRRs), focally enriched regions (FERs) and normal regions (NORs) in P2 and P5, respectively. Each dot represents one region. *P*, Wilcoxon test, two-tailed. *n*, number of tissue regions for TRRs, FERs, and NORs, respectively. Boxplots extend from the 25th to the 75th percentile, horizontal bars represent the median, and whiskers extend from $-1.5 \times IQR$ to $+1.5 \times IQR$ from the closest quartile, where IQR is the inter-quartile range. **k, l** Spatial distribution of the cells belonging to the five most localized pseudo-diploid subclones (C) identified in sample P2 and P5, respectively, and corresponding histopathological classification of each region, as in Fig. 1c, d. *n*, number of cells in each subclone. The anatomical orientation of the maps is as in Fig. 1c–j. A link to the Source Data for this figure is provided in the Data Availability statement.

whereas high values mark subclones localized to multiple regions. In agreement with the spatial maps, most of the subclones displayed intermediate entropy values and were distributed across many regions, whereas a small fraction of subclones were associated with either very low or high entropy values (Fig. 2g, h). Visual inspection of the copy number profiles of the cells in the five most localized subclones revealed that they were typically similar pseudo-diploid cells within the same subclone, as expected (Supplementary Figs. 19 and 20). Notably, in the P2 sample, the mean entropy value of TRRs was significantly lower ($P = 0.0031$, Wilcoxon test, two-tailed) compared to NORs in the same sample (Fig. 2i), indicating that highly localized pseudo-diploid subclones are concentrated in tumor-rich regions. A similar trend was observed in sample P5, although the difference between TRRs and NORs was not statistically significant, most likely due to an outlier region (Row 2, Column 6) in which only a few cells (17) passed quality control (Fig. 2j). In line with these findings, in both samples, the regions harboring all or most of the cells belonging to the most localized subclones typically overlapped with either TRRs or FERs (Fig. 2k, l). The only exception was subclone 32 in P5, which was found in five NORs adjacent to TRRs or FERs (Fig. 2l). In contrast to pseudo-diploid cells, aneuploid cells were not enriched inside TRRs and did not form clearly distinguishable subclones, even though they could be separated into distinct large clusters using uniform manifold approximation and projection (UMAP)[22] (Supplementary Figs. 21 and 22). Altogether, these results indicate that a subset of clonally related pseudo-diploid cells is highly localized within tumor-rich regions, suggesting that specific alterations in these cells might drive tumorigenesis.

## Spatial mapping of pseudo-diploid cells

To further investigate the spatial distribution of pseudo-diploid cells and validate scCUTseq, we performed DNA fluorescence in situ hybridization (FISH) to visualize the distribution of a pseudo-diploid cell subclone featuring a monoallelic sub-chromosomal arm deletion on chr13 detected at high frequency by both scCUTseq (77.3% of the cells retained by our random forest classifier) and ACT (72.2%) inside one TRR (Row 4, Column 3) of one of the other four prostate samples (P6) collected in this study (Fig. 3a, b). To this end, we leveraged the iFISH pipeline that we previously developed[23] and generated three DNA FISH probes in three colors targeting, respectively, the middle part of the deleted chromosomal region as well as one upstream and one downstream flanking region (Fig. 3c, Supplementary Data 3, and Methods). Using these probes, wild-type chr13 copies should appear as clusters of three close-by fluorescence spots—one spot for each of the three probe colors—whereas clusters containing only two signals originating from the two flanking probes should mark the deleted chr13 copies. Indeed, whole-slide widefield imaging (25X magnification) followed by deconvolution with our open-source software Deconwolf[24] of the top tissue section obtained from the same TRR profiled by scCUTseq and ACT detected both types of clusters throughout the entire tissue section, allowing to pinpoint nuclei carrying the chr13 deletion (Fig. 3d, e). Next, we quantified FISH signals across 51 fields of view (FOVs, 100X magnification), revealing that, on average, the dot counts corresponding to the probe marking the deletion were only ~75% of the downstream probe dot counts used as reference, compared to ~100% for the upstream probe signals (Fig. 3f and Methods). Because the deletion is monoallelic, this can be interpreted as ~50% of the cells in the FOVs examined carrying the deletion, which is consistent with the fraction (~50%) of the same FOVs overlapping with tissue section areas classified as 'tumor' by a pathologist (Supplementary Fig. 23). Indeed, we found a very strong positive correlation (Pearson's correlation coefficient, PCC: 0.89) between the number of cells in a given FOV, which carried the deletion, and the area in same FOV overlapping with annotated tumor regions (Fig. 3g), indicating that these cells represent bona fide tumor cells. These results further validate scCUTseq with an orthogonal method, and highlight the power of iFISH combined with Deconwolf[24] to quantitatively detect subclonal deletions across large tissue regions, which could be harnessed in future diagnostic applications.

## Loss of tumor suppressors in highly localized pseudo-diploid clones

The observation that highly localized pseudo-diploid cell subclones overlap with TRRs or FERs and typically carry deletions of one or few sub-chromosomal regions led us to hypothesize that they might represent tumor-initiating subclones in which one or more tumor-suppressor genes (TSGs) have been lost. To test this hypothesis, we focused on pseudo-diploid subclones detected exclusively within TRRs or FERs and identified 7 TRR/FER-specific pseudo-diploid subclones in P2 and 2 in P5, all of which featured deletions only (Fig. 4a and Supplementary Fig. 24a). In P2, two groups of phylogenetically related TRR/FER-specific pseudo-diploid subclones (Group 1: subclones 53–56; Group 2: subclones 66 and 69) localized to the anterior-left and posterior-right poles of the prostate midsection, respectively, and carried two deletions affecting common regions on chr6 (see single asterisks) and chr13 (double asterisks), respectively, whereas one subclone (65) was detected in four regions along the posterior margin and only featured a chr12 deletion (Fig. 4b–i). In P5, both TRR/FER-specific pseudo-diploid subclones were localized in the main tumor area in the anterior-right pole of the prostate midsection and carried a chr8 p-arm deletion also found in three (53–55) of the Group 1 subclones in P2 (Fig. 4c–e and Supplementary Fig. 24b–d).

Next, we checked how many genes annotated as TSGs in the Catalogue of Somatic Mutations in Cancer (COSMIC)[25] were lost in TRR/FER-specific pseudo-diploid subclones detected exclusively within TRRs or FERs. We identified 42 and 16 TSGs monoallelically deleted in P2 and P5, respectively, including several TSGs deleted in both tumor samples (*ARHGEF10*, *CDKN1B*, *ETNK1*, *ETV6*, *LEPROTL1*, *NRG1*, *PTPN6*, and *WRN*) and many TSGs co-deleted in the same subclone as well as across multiple subclones (Fig. 4a, Supplementary Fig. 24a, and Supplementary Data 4). In P2, all the six subclones belonging to Group 1 and 2 had 8 TSGs (*BRCA2*, *CCNC*, *CDX2*, *FOXO1*, *FOXO3*, *LATS2*, *PRDM1*, *RB1*) co-deleted in the same cells (Fig. 4a). These TSGs map to two regions on chr6 and chr13 q-arms that are frequently deleted in prostate adenocarcinomas in The Cancer Gene Atlas (Fig. 4j and Supplementary Fig. 24e). Of note, two of these genes

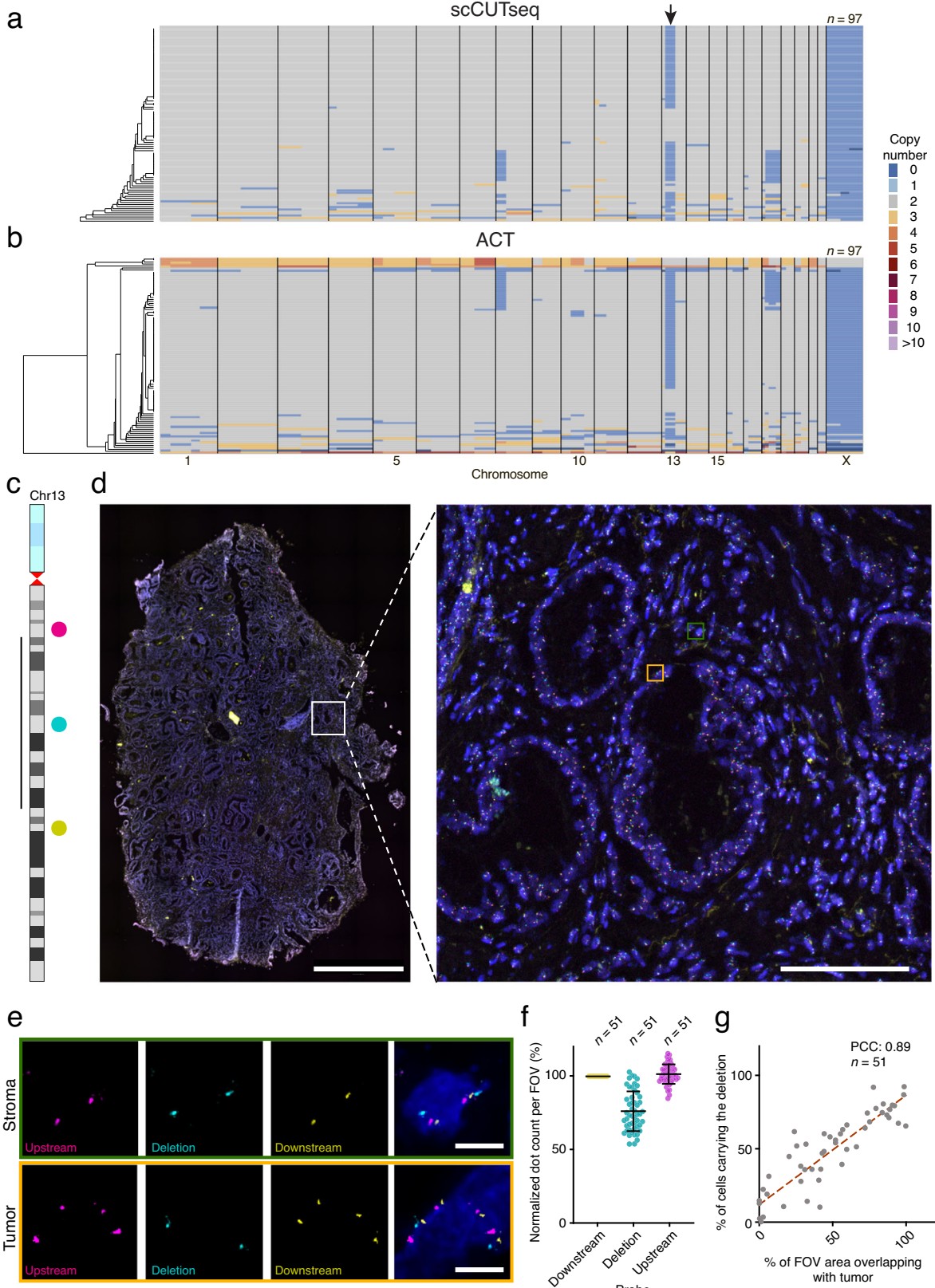

(*FOXO1* and *FOXO3*) encode members of the Forkhead transcription factor family, which includes the *FOXA1* gene frequently mutated in prostate cancers[26] and have been previously functionally annotated as context-dependent TSGs or oncogenes[27].

Prompted by these findings, we then mapped all the cells harboring a co-deletion of the aforementioned eight TSGs. Most of these cells localized in FERs and TRRs, with the highest concentration in one FER (Row 1, Column 9) and two TRRs (Row 1, Column 8; Row 4, Column 2) corresponding to the regions in sample P2 with the highest fraction of TRR/FER-specific pseudo-diploid subclones (Fig. 4k). Notably, the same cells accounted for the highest proportion of pseudo-diploid as well as of all the cells yielding high-quality copy number profiles from

**Fig. 3 | Validation and spatial mapping of pseudo-diploid cells identified by scCUTseq. a, b** Single-cell copy number profiles (500 kb resolution) obtained by applying scCUTseq (a) or Acoustic Cell Tagmentation (ACT) (b) to nuclei extracted from a single tumor-rich region in prostate sample P6. *n*, number of cells. The arrow indicates a subclonal deletion on chr13 q-arm detected by both scCUTseq and ACT. **c** Chr13 ideogram showing the location of the three DNA FISH probes (colored dots) used to detect the chr13 deletion spanning the region indicated by the vertical bar on the left and corresponding to the deletion marked by the arrows in (**a, b**). **d** Whole-slide low-magnification (25X) imaging after DNA FISH was performed using the probes shown in (**c**). The white squared region is magnified on the right. Scale bars, 1 mm (left) and 100 μm (right). **e** Examples of nuclei with both chr13 copies intact (stroma) or with one of the two copies carrying the deletion shown in (**c**) (tumor). White arrows indicate missing cyan dots, which correspond to the FISH probe targeting the deleted region. The nuclei shown are 100X magnification

zoom-in views of the same-color squared regions in (**d**). Scale bars, 5 μm. **f** Quantification of the number of fluorescent dots corresponding to each of the DNA FISH probes in (**c**), across 51 (*n*) fields of view (FOVs) imaged at high magnification (100X) in the whole-slide image shown in (**d**). The number of dots in each color was normalized to the number of yellow dots, corresponding to the probe downstream of the deleted region, as shown in (**c**). Each dot represents one FOV. Error bars span from −1.5 × IQR to +1.5 × IQR from the closest quartile. IQR, interquartile range. Horizontal bar, median. **g** Correlation between the fraction of cells carrying the deletion in each of the FOVs (*n*) analyzed in (**f**), and the proportion of the area of the corresponding FOVs that overlaps with regions annotated as tumor (see Supplementary Fig. 23). Dashed red line: linear regression. PCC, Pearson's correlation coefficient. A link to the Source Data for this figure is provided in the Data Availability statement.

the same regions (including diploid and aneuploid cells) (Fig. 4l, m). These results suggest that monoallelic loss of the above-listed eight TSGs (possibly concomitant with inactivation of the other allele) might drive early carcinogenesis in the prostate. Indeed, analysis of TCGA prostate adenocarcinomas (PRADs) revealed that 28.7% of all PRADs—including many Grade Group 2–3 PRADs corresponding to the same Gleason score (7) assigned to sample P2—carry a loss of one or multiple of those TSGs. From these genes, *RB1* and *FOXO1* are the first and second most frequently deleted TSGs, and *CCNC-FOXO3-PRDM1* and *CCNC-FOXO1-FOXO3-PRDM1-RB1* the first and second most frequent co-deletion combination occurring in these tumors, respectively (Fig. 4n). Among the genes frequently deleted in TCGA PRADs we also found *FOXP1*, another member of the Forkhead transcription factor family. However, this gene is almost never co-deleted with the other TSGs in TCGA PRADs (Fig. 4n), indicating that certain TSG deletions might be mutually exclusive. Accordingly, we only detected 3 pseudo-diploid cells with a *BRCA2-FOXO1-FOXO3-FOXP1-RB1* co-deletion, two of which localized in TRRs (Row 1, Column 8; Row 4, Column 2) containing a high density of cells carrying the 8 TSGs co-deletion described above. We did not find deletions affecting *NKX3-1*, *PTEN* and *TP53* genes, which are frequently deleted in TCGA PRADs. However, this might be due to the very limited sample size of our proof-of-concept study.

Lastly, we wondered whether the TRRs and FERs in the two prostate samples that we thoroughly profiled by scCUTseq also harbor mutations in genes previously implicated in prostate cancer. We identified mutations in several genes frequently mutated in PRADs[28], including *FOXA1*, *FOXP1*, *LRP1B*, *SPTA1*, and *SPOP* in sample P2, and *FOXA1* and *LRP1B* in sample P5 (Supplementary Fig. 24f, g and Supplementary Data 5). In sample P2, *LRP1B*, *SPTA1*, and *SPOP* mutations were found in the same TRRs/FERs in which we detected the highest density of pseudo-diploid cells carrying the 8-TSG co-deletion described above, as well as in few other TRRs/FERs (Fig. 4o–q). *FOXA1* mutations were found in two TRRs as well as in two NORs, whereas *FOXP1* was mutated only in one NOR along the anterior margin of the prostate midsection (Fig. 4r, s). In sample P5, only *LRP1B* mutations were detected in one of the TRRs in the anterior-right pole of the prostate midsection, whereas *FOXA1* mutations were found in a few FERs/NORs along the posterior margin (Supplementary Fig. 24h, i). Of note, because these mutations were not detected in the same cells profiled by scCUTseq, it is not possible to ascertain whether they co-existed with the TSG co-deletions identified in pseudo-diploid cells. However, the concomitant finding of these events within the same prostate regions suggests that they might cooperatively act to initiate or promote carcinogenesis in prostate glands marked by the ubiquitous presence of cells with SCNAs.

## Discussion

Leveraging scCUTseq to perform spatially resolved single-cell profiling of SCNAs across normal and tumor regions in two prostates from two

patients diagnosed with localized early-stage prostate cancer, we have uncovered the existence of two major types of genomically unstable cells—pseudo-diploid and aneuploid cells—characterized by profoundly different SCNA profiles most likely reflecting a different mechanism of origin. Pseudo-diploid cells carry a few (typically, between 1 and 7) (sub-)chromosome arm alterations, predominantly consisting of deletions. In contrast, aneuploid cells carry genome-wide whole-chromosome copy number changes and are reminiscent of 'hopeful monsters', which were previously shown to emerge continuously in propagating colorectal cancer organoid cultures, as a result of catastrophic mitotic errors[7]. We detected pseudo-diploid and aneuploid cells throughout all the prostate regions examined, including normal regions distant from tumor regions, suggesting that they originate from widespread genome instability processes that are active throughout the prostate gland. Nonetheless, the fact that pseudo-diploid cells—but not aneuploid cells—can be clustered in different subclones with similar SCNA profiles indicates that SCNAs in these cells are still compatible with cell proliferation and thus can be propagated, unlike the highly aberrant karyotypes of aneuploid cells, representing singular events that cannot be propagated.

In addition to widespread aneuploid and pseudo-diploid cells, our study reveals the existence of a sub-population of pseudo-diploid cells that are highly localized within tumor-rich regions and carry sub-chromosomal deletions resulting in concurrent loss of multiple tumor-suppressor genes, thus potentially representing tumor-initiating/driving cells in these regions. In support of this hypothesis, in one (P2) of the two prostate samples that we extensively profiled by scCUTseq, we found that tumor regions were enriched in multiple phylogenetically related subclones with a monoallelic loss of multiple TSGs, including *BRCA2*, *CCNC*, *CDX2*, *FOXO1*, *FOXO3*, *LATS2*, *PRDM1*, and *RB1*. These TSGs are frequently deleted alone or in various combinations in prostate adenocarcinomas (PRADs) in TCGA and *RB1* has been shown to be frequently deleted with *TP53* and *PTEN* in localized prostate cancers[29]. Of note, we did not uncover deletions in the latter two genes even though these are the most frequently deleted genes in TCGA PRADs, most likely reflecting the very limited number of patient samples examined in this study. Furthermore, *FOXO1*, *LATS2*, and *RB1* have been shown to act as tumor suppressor genes in various prostate cancer models[30–32]. Notably, *FOXO1* and *FOXO3* are members of the large family of Forkhead transcription factors involved in multiple physiological and pathological processes[27], which also includes *FOXA1* and *FOXP1* genes that are mutated (*FOXA1*) or deleted (*FOXP1*) in a subset of PRADs and have been previously implicated in prostate carcinogenesis[33–35]. We speculate that disruption of Forkhead transcription factor networks, coupled with dosage effects on multiple TSGs, represents a tumor-initiating event independently occurring at multiple, spatially distinct locations in the prostate gland, in a subset of patients with prostate cancer. Future studies leveraging emerging spatially resolved genomic technologies to profile SCNAs and

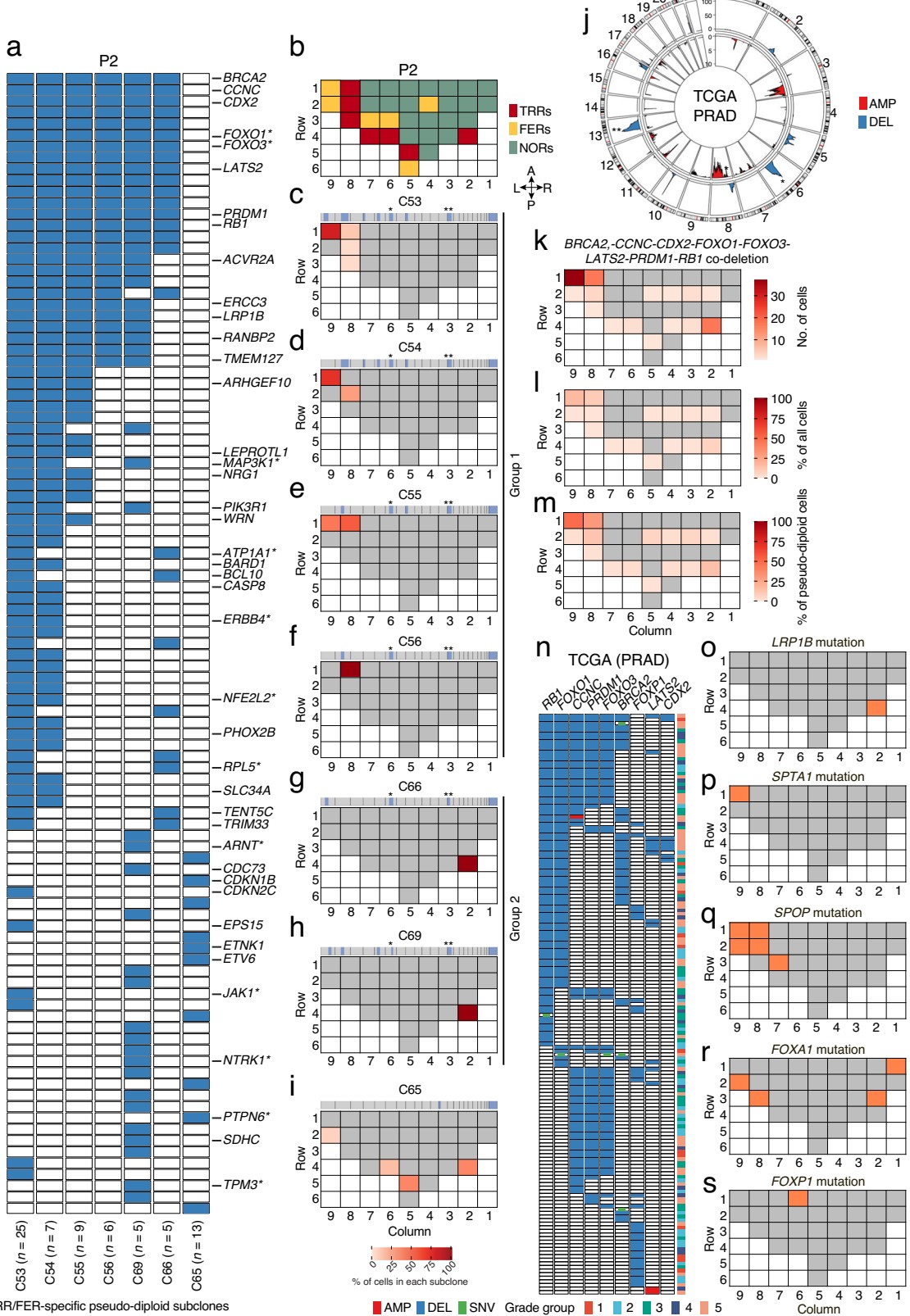

mutations at high spatial resolution across large sample cohorts will enable addressing this hypothesis.

The Gleason grading system represents the main tool used for predicting the prognosis and guiding the therapy of patients affected by prostate cancer[36]. This system, however, is blind to the genomic makeup of the cells analyzed and to spatial genetic heterogeneity across different regions of the prostate, which likely encode important prognostic information. Instead, the combined spatially resolved single-cell SCNA profiling and targeted deep sequencing approach that we have adopted here can capture genetic diversity and spatial heterogeneity throughout the prostate in an unbiased manner, providing a quantitative portray of the type, relative proportions, and spatial

**Fig. 4 | Tumor suppressor genes lost in highly localized pseudo-diploid sub-clones are putative drivers of early prostate tumorigenesis. a** OncoPrint plot showing genes classified as tumor-suppressor genes (TSGs) in the Catalogue of Somatic Mutations in Cancer (COSMIC) that were deleted (blue rectangles) in pseudo-diploid subclones (C) localized exclusively in tumor-rich regions (TRRs) or focally enriched regions (FERs) in prostate sample P2. *n*, number of pseudo-diploid cells in each subclone. Asterisks indicate genes annotated both as TSGs and as oncogenes (context-dependent TSGs) in COSMIC. **b** As in Fig. 1c. **c–i** Spatial distribution of the seven pseudo-diploid subclones (C) shown in (**a**). The median copy number profile of each subclone is shown on top of the corresponding tissue map. Asterisks indicate regions on chr6 (*) and chr13 (**) that are deleted in many subclones. **j** Circos plot showing chromosomal regions frequently amplified (AMP) or deleted (DEL) (identified using GISTIC[63]) across 492 (*n*) prostate adenocarcinoma

(PRAD) samples in The Cancer Genome Atlas (TCGA). Y-axis, $-\log_{10}$(q-value). The asterisks mark the regions on chr6 and 13 that were found deleted in multiple pseudo-diploid subclones, as shown in (**c–h**). **k–m** Spatial distribution of the pseudo-diploid cells carrying a co-deletion of the indicated eight TSGs in sample P2. **n** OncoPrint plot displaying TCGA PRAD samples with an amplification (AMP), deletion (DEL) or mutation (SNV) of the eight TSGs shown in (**k–m**) plus *FOXP1*. Each row in the plot corresponds to one TCGA PRAD sample and the corresponding Grade Group is shown in the colorbar on the right. **o–s** Schematic maps showing where the indicated genes were found mutated (non-synonymous SNVs) in sample P2. Gray cells indicate tissue regions in which the indicated gene was not altered. White cells indicate absence of tissue. The anatomical orientation of all the maps in (**c–i**) and (**o–s**) is as in (**b**). A link to the Source Data for this figure is provided in the Data Availability statement.

distribution of cells with different karyotypes as well as of potential cancer driver mutations. We propose using this approach to derive a 'Genomic Diversity Index', which could then be integrated with the Gleason score to improve prognostication for patients affected by prostate cancer.

The scCUTseq method that we have developed and applied here is versatile and cost-effective and is especially suited for profiling SCNAs in single nuclei extracted from patient-derived samples, since the nuclei can be first stabilized by fixation and stored up to several weeks before being sorted and used for preparing sequencing libraries. The use of fixed nuclei was not demonstrated before in tagmentation-based scDNA-seq methods such as DLP + [37], DNRT[38], and ACT, although we now show that ACT can also be applied to fixed samples, similarly to scCUTseq. Of note, the tagmentation-based scDNA-seq method Arcwell, which uses nuclei extracted from archival formalin-fixed paraffin-embedded (FFPE) tissues, was recently described and applied to study the clonal structure of ductal in situ carcinoma of the breast[39]. Although in this study we have not tested scCUTseq on FFPE samples, we expect that our method should also be applicable to these samples since it builds on the CUTseq method, which we previously specifically developed for FFPE samples[16].

One inherent limitation of scCUTseq is the sparse breadth of genome coverage that can be achieved (typically, less than 2%) when single restriction enzymes are used, as in this study. This limits the theoretical maximum genomic resolution that can be achieved (as we previously extensively discussed for bulk CUTseq[16]) and precludes the possibility of determining B-allele frequencies and hence obtaining phased copy number profiles. However, as we have shown in this study, scCUTseq detects very similar copy number profiles and cell populations as tagmentation-based ACT, indicating that the use of restriction enzymes is fully compatible with single-cell SCNA profiling in tumors and other tissues. Of note, another scDNA-seq method named Karyo-Seq also leverages restriction enzymes and was used to profile SCNAs in patient-derived tumor organoids[40, 41].

In conclusion, our study provides an organ-wide, spatially resolved single-cell map of SCNAs in prostate samples with focal adenocarcinoma and reveals a previously undetected class of genomically unstable cells (pseudo-diploid) carrying recurrent (sub-)chromosomal deletions representing putative tumor-initiating cells. Future studies combining concurrent genome and transcriptome sequencing from the same cell with spatial resolution will be pivotal to characterize this cell population in depth.

## Methods
### Ethical regulation statement
All research complies with the ethical regulations and was approved by the relevant Ethical Committees. The collection of prostate samples was approved by the Regional Ethics Committee of Sweden, ethical permit: 2018/1003-31. The collection of brain and skeletal muscle samples was approved by the Regional Ethics Committee of Sweden,

ethical permit: 2010/313-31/3. The collection of breast cancer samples was approved by the Candiolo Cancer Institute Ethical Committee, ethical permit: "Profiling", 001-IRCC-00IIS-10. All patient-derived samples were collected and used in conformity to the permits. None of the donors received compensation. Written informed consent was obtained for the donors of the prostate and breast cancer samples. Informed consent was not specifically obtained for the present study for the brain and skeletal muscle samples, since the study was not conceived at the time of collection. However, ethical approval was obtained for the use of these tissues. Sex and/or gender was determined based on self-reporting but was not taken into consideration during the analysis. For more information, please refer to the Reporting Summary.

### Experimental methods
#### Samples
**Cell lines.** We purchased IMR90, SKBR3 and MCF10A cell lines from the American Tissue Culture Collection (ATCC, cat. no. CCL-186, HTB-30, and CRL-10317, respectively) and Drosophila S2 cells from Gibco (cat. no. R69007). TK6 cells were previously described[42]. None of these cell lines is registered in the International Cell Line Authentication Committee (ICLAC) database of misidentified cell lines. We periodically tested all the cell lines for mycoplasma contamination using the MycoAlert Mycoplasma Detection Kit (Lonza, cat. no. LT07-118) and consistently obtained negative test results. Culturing conditions were as follows: (i) IMR90 cells: Eagle's Minimum Essential Medium (EMEM) (Sigma, cat. no. M5650) supplemented with 10% heat-inactivated fetal bovine serum (FBS; Sigma, cat. no. F9665), 2 mM L-glutamine (Sigma, cat. no. 59202 C), and 1% non-essential amino acids (Gibco, cat. no. 11140035); (ii) SKBR3 cells: McCoy's 5 A Medium (Sigma, cat. no. M9309) supplemented with 10% heat inactivated FBS (Sigma, cat. no. F9665); (iii) MCF10A cells: Mammary Epithelial cell Growth Medium (MEGM) (Lonza, cat. no. CC-3150) supplemented with 100 ng/ml cholera toxin (Sigma, cat. no. C8052) according to the ATCC guidelines; (iv) S2 cells: Schneider's medium (Gibco, cat. no. 21720024) supplemented with 10% heat inactivated FBS (Sigma, cat. no. F9665); (v) TK6 cells: Roswell Park Memorial Institute Medium (GIBCO, cat. no. 11875093) supplemented with 5% horse serum (GIBCO, cat. no. 11510516), 1 mM sodium pyruvate (GIBCO, cat. no. 11360070), 100 U/mL penicillin-streptomycin (GIBCO, cat. no. 15140122) and 2 mM L-glutamine (GIBCO, cat. no. 25030081). We grew all the cell lines at 37 °C in 5% CO2 air, except for S2 cells, which were grown at 28 °C in a non-humidified, ambient air-regulated incubator.

**Prostate samples.** We prospectively collected six prostatectomy samples (P1-P6) from male patients aged 43–65 and operated for localized prostate cancer by endoscopic surgery at the Södersjukhuset Hospital in Stockholm, Sweden. From the same patients, we also collected peripheral blood for gDNA extraction. The main pathological characteristics of each sample are summarized in Supplementary

Table 1. For each tumor, we obtained a ~ 0.5 cm thick transversal mid-section of the prostate and then immediately cut it into multiple ~0.5×0.5×0.5 cm³ tissue blocks. We embedded each block in Tissue-Tek OCT compound (VWR, cat. no. 00411243), snap-froze it in iso-pentane, and stored it at −80 °C. We stained one section from the top and one from the bottom of each block with hematoxylin-eosin (H&E), and then scanned each section using a Hamamatsu Nano Zoomer-XR Digital slide scanner. Two board-certified pathologists at Qilu Hospital of Shandon University, China independently evaluated each tissue section and performed Gleason grading following the International Society of Urological Pathology (ISUP) 2014/World Health Organization 2016 modified system[43]. Accordingly, regions containing more than 50% of tumor cells were classified as tumor-rich regions, whereas regions containing 10-50% of tumor cells were labeled as focally enriched regions. All the prostate and breast cancer samples described above are unique biological samples the use of which is restricted to this study, and thus cannot be distributed to other researchers.

**Brain and skeletal muscle samples.** We retrieved fresh frozen human prefrontal cortex tissue and skeletal muscle samples from two donors (1 male and 1 female, aged 45−50. Also see Supplementary Table 2) that had been procured through the KI Donatum Tissue Collection program, a collaboration between the Swedish National Board of Forensic Medicine and the Department of Oncology-Pathology at Karolinska Institutet.

**Breast cancer samples.** We retrieved two frozen Luminal B-like breast cancer specimens (here labeled B1 and B2) at the Pathology Unit of the Candiolo Cancer Institute (Italy). The samples had been previously collected from two female patients, aged 55−65, and stored in the frame of a prospective study for the identification of molecular profiles conferring resistance to selected target therapies in oncological patients. Following excision in the surgical theater, the samples were immediately placed in vacuum and stored at + 4 °C until further processing. ~1×1×0.5 cm³ tissue blocks were embedded in Tissue-Tek OCT compound (VWR, cat. no. 00411243), snap-frozen in isopentane, and stored at −80 °C.

**scCUTseq.** A detailed step-by-sep scCUTseq protocol is available at Protocol Exchange.

**Cells and nuclei preparation.** We prepared cultured cells for standard MALBAC as described in the Supplementary Methods. For prostate samples, we cut each tissue block into halves and kept each half in DNA/RNA Shield at −80 °C until further processing. We prepared single nuclei suspensions from breast and prostate samples following the Tapestri frozen tissue nuclei extraction protocol (https://missionbio.com). Briefly, we first prepared a spermine solution composed of 3.4 mM sodium citrate tribasic dihydrate (Sigma, cat. no. C8532)/1.5 mM spermine tetrahydrochloride (Sigma, cat. no. S1141)/0.5 mM tris (hydroxymethyl) aminomethane (Sigma, cat. no. 252859)/0.1% v/v IGEPAL CA-630 (Sigma, cat. no. I8896) and stored it at +4 °C. We retrieved the frozen breast cancer or prostate cancer tissue blocks and kept them on dry ice until further processing. We then transferred one sample at a time into a pre-chilled Petri dish placed onto a dry-ice pad, added 200 µL of tissue lysis solution containing spermine solution (pH 7.6)/0.03 mg/mL trypsin-EDTA (0.25%), phenol red (Thermo Fisher scientific, cat. no. 25200072)/0.1 mg/mL collagenase (Worthington, cat. no. CLS-7 LS005332)/1 mg/mL Dispase II (Gibco, cat. no. 17105-041) on top of the tissue block and incubated it until the tissue lysis solution had frozen ( ~ 3 min). We minced the sample thoroughly with two pre-chilled sterile scalpels and then transferred it to room temperature, while continuing mincing until all the tissue fragments could flow through a 1 mL pipette tip without clogging it. We added additional 1.8 mL of tissue lysis solution to rinse the dish, transferred

all the tissue fragments from the Petri dish into a clean 5 mL low-binding tube (Eppendorf, cat. no. 0030108.310), and rotated the tube at 20 rpm at room temperature for 15 min. We stopped the lysis by adding 2 mL of stop solution containing spermine solution (pH 7.6)/0.5 mg/mL trypsin inhibitor from chicken egg white, type II-O (Sigma, cat. no. T9253)/ 0.1 mg/mL ribonuclease A from bovine pancreas, type I-A (Sigma, cat. no. R4875-100mg) and mixed the solution by gently inverting the sample. Next, we filtered the solution through a 50 µm cell strainer and collected the flowthrough in a clean 5 mL low-binding tube. We centrifuged the flowthrough at $300 \times g$ for 5 min at room temperature and discarded the supernatant. To fix the nuclei, we resuspended them in 400 µL of a nuclei fixation solution containing 66% v/v methanol/33% v/v acetic acid pre-chilled at +4 °C and thoroughly pipetted the nuclei suspension up and down. After 15 min incubation on ice, we centrifuged the sample at $300 \times g$ for 5 min at room temperature to collect the fixed nuclei. We discarded the supernatant and resuspended the nuclei in 1X PBS/5 mM EDTA/0.05% NaN₃ and filtered the nuclei through a 20 µm cell strainer. We have successfully used fixed nuclei prepared in this manner, that were kept at 4 °C for several weeks up to two months. We extracted gDNA and RNA from the tissue blocks simultaneously using the Quick-DNA/RNA Microprep Plus Kit (ZYMO Research, cat. no. D7005). For DNA and RNA purification, we first homogenized the tissues that were stored in 1×DNA/RNA Shield following the instructions for Tough-to-Lyse samples using Biomedical FastPrep Homogenizer (MP Bio). We then performed the DNA and RNA purification steps following the manufacturer's instructions. We checked DNA/RNA quality and quantity with NanoPhotometer (Implen).

To isolate neuronal cell nuclei from human prefrontal cortex tissue, we thawed and homogenized ~1 g of tissue in 10 mL of an ice-cold lysis buffer containing 10 mM Tris-HCl pH 8.0/0.32 M sucrose/5 mM CaCl₂/3 mM magnesium acetate/0.2 mM EDTA/0.1% Triton X-100/1 mM DTT using a glass Douncer. We added 20 mL of ice-cold 1.8 M sucrose solution to the homogenate and layered the mix onto a cushion of 10 mL 1.8 M sucrose solution followed by centrifugation at $30{,}000 \times g$ for 2 h at 4 °C. We carefully aspirated the supernatant and resuspended the pellet in 1.5 mL of a nuclei storage buffer containing 10 mM Tris-HCl pH 7.2/15% sucrose/70 mM KCl/2 mM MgCl₂/1.5 mM spermine tetrahydrochloride. We fixed the isolated nuclei in 4% methanol-free formaldehyde (Thermo Fisher Scientific, cat. no. 28908) for 10 min at room temperature and quenched the reaction in 1X PBS pH 7.4/125 mM glycine for 5 min at room temperature. We centrifuged the samples at $500 \times g$ for 10 min at 4 °C and washed the nuclei once in 1X PBS/10 mM EDTA pH 7.4. To isolate neuronal cell nuclei, we incubated the nuclei suspension on ice for 30 min with an anti-NeuN antibody (Alexa647-conjugate, clone A60, Merck, cat. no. MAB377) diluted 1:500 in 1X PBS/10 mM EDTA pH 7.4 followed by filtration of the nuclei suspension through a 30 µm pre-separation filter (Miltenyi Biotec, cat. no. 130-041-407) to remove large particles and nuclei aggregates. Lastly, we stained the nuclei with Hoechst 33342 (Thermo Fisher Scientific, cat. no. 62249) and proceeded to sort single nuclei as described below.

To isolate nuclei from human skeletal muscle tissue, we minced the tissue samples with a disposable scalpel before homogenizing them. We homogenized ~0.5 g of tissue in 10 mL of an ice-cold homogenization buffer containing 10 mM HEPES pH7.4/60 mM KCl/2 mM EDTA/0.5 mM EGTA/300 mM sucrose/0.5 mM spermidine/0.15 mM spermine using a T25 Ultra Turrax (IKA) device set at 25,000 rpm for 10 sec. We added 10 mL of ice-cold homogenization buffer supplemented with 1% BSA, followed by homogenization using a glass Douncer, filtration through a 100 µm cell strainer (Falcon BD, cat. no. 352360) to remove debris and large aggregates, and centrifugation at $4500 \times g$ for 10 min at 4 °C. We resuspended the nuclei pellet in 1.25 mL of 10 mM HEPES pH 7.4/60 mM KCl/0.1 mM EDTA/0.1 mM EGTA/300 mM sucrose/0.5 mM spermidine/0.15 mM spermine/0.5%

bovine serum albumin pre-mixed with 0.75 mL of a Percoll stock solution containing 80% Percoll (Sigma Aldrich, cat. no. P1644)/10 mM HEPES pH 7.4/60 mM KCl/0.1 mM EDTA/0.1 mM EGTA/300 mM sucrose. We centrifuged the samples at $16{,}900 \times g$ for 30 min at 4 °C followed by removal of the compressed myofibrillar layer from the top of the suspension. We then collected the floating nuclear layer and transferred it into 1 mL of a nuclei storage buffer containing 10 mM Tris-HCl pH 7.2/15% sucrose/70 mM KCl/2 mM MgCl$_2$/1.5 mM spermine tetrahydrochloride, followed by centrifugation at $700 \times g$ for 5 min at 4 °C. We resuspended the nuclei pellet in 1X PBS/3% BSA and added methanol-free formaldehyde (Thermo Fisher Scientific, cat. no. 28908) to reach a final concentration of 4%, followed by incubation for 10 min at room temperature. We quenched the reaction in 1X PBS pH 7.4/125 mM glycine for 5 min at room temperature, centrifuged the samples at $500 \times g$ for 10 min at 4 °C, and washed the nuclei once in 1X PBS pH 7.4/10 mM EDTA. Lastly, we stained the nuclei with Hoechst 33342 (Thermo Fisher Scientific, cat. no. 62249) and proceeded to sort single nuclei.

**Preparation of CUTseq adapters.** We used either the 96 CUTseq adapters previously described[16] or 384 adapters that we newly designed. The sequences of all the oligonucleotides (oligos) used are available in Supplementary Data 6. We purchased the oligos from Integrated DNA Technologies (IDT) as standard desalted oligos at 100 μM in Nuclease-Free Water in 96-well plates. In each well of a 96- (for 96 adapters) or 384-well plate (for 384 adapters), we dispensed 5 μL of a sense oligo and then added 40 μL of phosphorylation reaction mix containing 1 μL of T4 Polynucleotide Kinase (PNK; NEB, cat. no. M0201S), 5 μL of T4 PNK buffer (NEB, cat. no. M0201S) and 5 μL of 10 mM ATP (Thermo Fisher Scientific, cat. no. PV3227) in Nuclease-Free Water and incubated at 37 °C for 30 min followed by inactivation at 65 °C for 20 min. Then, we added 5 μL of the corresponding antisense oligo to each well and annealed the complementary sense and antisense oligos by incubating the plates at 95 °C for 5 min, followed by cooling down to 25 °C over a period of 45 min in a PCR thermocycler. Afterwards, we diluted the annealed adapters to 33 nM in Nuclease-Free Water and stored them at −20 °C. Adapter dilutions stored in this way are stable for 6–8 months.

**Single nucleus sorting.** Before sorting, we manually dispensed 5 μL of Vapor-Lock (Qiagen, cat. no. 981611) into each well in the targeted region of a 384-well plate. We then transferred the samples into FACS-compatible tubes, added 2.46 ng/mL Hoechst 33342 (Thermo Fisher Scientific, cat. no. 62249) to them and incubated for 2 min on ice in darkness, while rotating. We sorted single nuclei in 96- or 384-well plates (one nucleus per well in 10 nL sorting volume) using the BD FACSJazz Cell Sorter or BD Aria III (BD Biosciences) based on forward and side scatter properties. After sorting, we immediately sealed the plates, spun them at 3,220×g for 5 min and stored them at −20 °C until further processing.

**Scaled down MALBAC.** We used the MALBAC kit (Yikon Genomics, cat. no. Y001A) for whole genome amplification and the I.DOT nanodispensing system (CELLINK) for dispensing all reagents in small volumes, to increase the efficiency of reactions while limiting the reagent costs per cell. For the cell lysis step, we prepared a lysis mix containing 30 nL of cell lysis buffer and 0.6 nL of cell lysis enzyme per cell. We then dispensed 30 nL of the lysis mix into each well and incubated the plates at 50 °C for 1 h and 80 °C for 10 min in a PCR thermocycler. For the pre-amplification step, we prepared a pre-amplification mix containing 150 nL of pre-amplification buffer and 5 nL of pre-amplification enzyme mix per cell. We then dispensed 150 nL of the pre-amplification mix into each well, and incubated the plates at 94 °C for 3 min to denature DNA, followed 8 cycles of quasi-linear amplification (20 °C for 40 sec; 30 °C for 40 sec; 40 °C

for 30 sec; 50 °C for 30 sec; 60 °C for 30 sec; 70 °C for 4 min; 95 °C for 20 sec; and 58 °C for 10 sec) in a PCR thermocycler. For the amplification step, we prepared an amplification mix containing 150 nL of amplification buffer, 4 nL of amplification enzyme mix and 11 nL of 4x SYBR Green (Thermo Fisher Scientific, cat. no. S7563) per cell, and kept the mix protected from light until we dispensed it. We dispensed 160 nL of the amplification mix into each well to reach a final volume of 350 nL per well and incubated the plates at 94 °C for 30 sec to denature DNA followed by 14 cycles of exponential amplification (94 °C for 20 sec; 58 °C for 30 sec; 72 °C for 3 min) in a PCR thermocycler. After each dispensing step, we shook the plates on a ThermoMixer at 1000 rpm, centrifuged the plates at 3220×g for 10 min, and placed them on ice until the next step.

**High-throughput CUTseq.** We performed CUTseq on the MALBAC products largely following the high-throughput CUTseq protocol based on the I.DOT system (Dispendix), which we described before[16]. Briefly, for the digestion step, we prepared a digestion mix containing 100 nL of NlaIII restriction enzyme (NEB, cat. no. R0125L) and 50 nL of CutSmart buffer (NEB, cat. no. R3104) per cell. We then dispensed 150 nL of the digestion mix into each well and incubated the plates at 37 °C for 30 min followed by 65 °C for 20 min to inactivate the enzyme. After digestion, we dispensed 300 nL of 33 nM scCUTseq adapters (prepared as described above) into each well (one barcode per cell), followed by 700 nL of a ligation mix containing 200 nL of T4 rapid DNA ligase (Thermo Fisher Scientific, cat. no. K1423), 300 nL of T4 ligase buffer (Thermo Fisher Scientific, cat. no. K1423), 120 nL of 10 mM ATP (Thermo Fisher Scientific, cat. no. PV3227), 30 nL of 50 mg/mL bovine serum albumin (Thermo Fisher Scientific, cat. no. AM2616), and 50 nL of Nuclease-Free Water (Thermo Fisher Scientific, cat. no. 4387936). We then incubated the plates at 22 °C for 30 min, after which we dispensed 5 μL of Nuclease-Free Water/33 mM EDTA (for a final concentration of 25 mM) into each well and pooled the contents of all the wells of each plate by placing the plates upside down onto a collection plate (NUNC, cat. no. 267060) covered with parafilm, and centrifuged them at $117 \times g$ for 1 min. We carefully transferred the collected solution into a 5 mL low-binding tube and then removed as much as possible of the Vapor-Lock. We then purified the pooled samples with a 1.2 v/v ratio of the sample and Agencourt Ampure XP bead suspension (Beckman Coulter, cat. no. A63881). After purification, we prepared 250 ng of DNA and sheared it using Covaris ME220 Focused-ultrasonicator with a target peak set at 200 base pairs (bp). To prepare sequencing libraries, we followed the CUTseq protocol[16], with the following minor modifications: (1) We increased the PCR volume to 200 μL and performed a split PCR reaction in 4-tubes strips; (2) We purified the final library with a 0.8 v/v ratio of sample and Agencourt Ampure XP bead suspension, and eluted the purified library in 50 μL of Nuclease-Free Water.

**Sequencing.** We first sequenced all scCUTseq libraries on NextSeq 500 (Illumina) in single-end mode using the NextSeq 500/550 High Output Kit (75 Cycles) (Illumina, cat. no. FC-404-2005) except for scCUTseq libraries from brain and skeletal muscle, which we sequenced on NextSeq 2000 (Illumina) in single-end mode using the NextSeq 2000 P2 Reagents (100 Cycles) kit (Illumina, cat. no. 20040559). For scCUTseq experiments on TK6 cells, breast cancer, and prostate samples, we sequenced high-quality libraries (pre-checked on Bioanalyzer) on NovaSeq 6000 (Illumina) in pair-end mode using the NovaSeq 6000 S4 flow cell (300 Cycles) kit (Illumina, cat. no. 20028312). See Supplementary Data 7 for a summary of sequencing statistics.

**scCUTseq validation by acoustic cell tagmentation (ACT).** To validate scCUTseq with a WGA-free scDNA-seq method, we adapted the protocol for Acoustic Cell Tagmentation (ACT)[18] for fixed nuclei and

implemented it on the nanodispensing device (I.DOT) used for scCUTseq. Briefly, we FACS-sorted single nuclei in 384-well plates prefilled with 5 µL of Vapor-Lock (Qiagen, cat. no. 981611) per well. For cell lysis, we adapted the same conditions used in Dip-C[44] and lysed each nucleus in 150 nL of lysis buffer containing 20 mM Tris pH8/ 20 mM NaCl/25 mM DTT/0.15% Triton X-100/1 mM EDTA/5 µg/mL Qiagen Protease (Qiagen, cat. no. 19157). After dispensing, we centrifuged the plate at $3220 \times g$ for 3 min, vortexed it at 1000 rpm for 1 min, and then again centrifuged it at $3220 \times g$ for 3 min. This was done after every dispensing step with I.DOT. For lysis, we incubated the plate at 50 °C for 1 h followed by heat inactivation at 70 °C for 15 min. To neutralize EDTA in the lysis buffer, we dispensed 50 nL of 4 mM MgCl$_2$ into each well and then vortexed and centrifuged the plate. For tagmentation, we dispensed 600 nL of tagmentation reaction mix containing Tagmentation DNA buffer (TD) and Amplicon Tagment Mix (ATM) at 2:1 v/v ratio (Nextera kit, Illumina, cat. no. FC-131-1096) into each well and performed tagmentation at 55 °C for 5 min followed by hold at 4 °C in a PCR thermocycler. To stop the reaction, we dispensed 200 nL of neutralization (NT) buffer into each well and incubated the plate for 5 min at room temperature. Lastly, we performed single nuclei indexing by dispensing 600 nL of Nextera PCR Master Mix (NPM) and 400 nL of a unique Nextera index pair (Illumina, cat. no. 20027213, 20027214, 20042666, 20042667) into each well. PCR settings were as follows: 72 °C for 3 min; 95 °C for 30 s; (98 °C for 10 s, 63 °C for 30 s, 72 °C for 30 s) for 16 cycles; 72 °C 5 min; hold +4 °C. Subsequently, we pooled the contents of all the wells of a 384-well plate together and purified the resulting library using AMPure XP beads (Beckman Coulter, cat. no. A63881) at 1.8 v/v ratio. We sequenced all ACT libraries on NextSeq 2000 (Illumina) in single-end mode using the NextSeq 1000/2000 P2 Reagents (100 Cycles) kit (Illumina, cat. no. 20046811). See Supplementary Data 7 for a summary of sequencing statistics.

**scCUTseq validation by DNA FISH.** To validate scCUTseq with an orthogonal method, we leveraged our iFISH pipeline for high-resolution DNA FISH[23] with several modifications to the original protocol specifically tailored for fresh-frozen tissue sections.

**Probe design and production.** We selected three regions of interest (ROIs) on chr13, one corresponding to the midpoint of a large deletion detected by both scCUTseq and ACT (chr13:54,000,000-55,000,000; hereafter "deletion probe") and two flanking regions, located ~3 Mb on either side of the deletion boundaries (chr13:29,000,000-30,000,000 and chr13:77,000,000-78,000,00; "upstream" and "downstream probe", respectively) (see Fig. 3c). We extracted all possible 40 nt target (T) sequences from each ROI and attributed each sequence a cost, depending on their level of off-target homology and intrinsic parameters such as melting temperature, delta free energy of secondary structures, and length of homopolymers. For each ROI, we selected the 6000 sequences providing the minimal combined individual cost (quality of each sequence) and pairwise cost (genomic distance between consecutive sequences). Lastly, we appended left (L) and right (R) 20 nt adapter sequences on the 5' and 3' side, respectively, of the T sequences. The sequences of all the oligos in each probe as well as the sequences of fluorescently labeled detection oligos are provided in Supplementary Data 3. We produced the probes using our iFISH pipeline[23] starting from synthetic oligopools (purchased from Twist Bioscience).

**Probe hybridization.** We obtained 10 µm-thick cryosections and directly transferred them onto microscope slides then stored at −80 °C until performing the experiments. For H&E staining, we first fixed the cryosections with cold acetone (prechilled at −20 °C) for 2 min followed by a brief wash in running water to remove acetone. We then stained the sections in Mayer's hematoxylin solution (VWR,

cat. no. 10047105) at 25 °C for 3 min. After a brief wash in running water, we transferred the sections into hot running water (preheated at 60 °C) for 10 min, followed by a short rinse in running water and staining in Eosin Y solution (Histolab, cat. no. 01650) for 1 min. We then dehydrated the sections through 95% ethanol and twice 100% ethanol, 10 sec each. Finally, after clearing the sections in xylene twice 5 min each, we mounted the sections with Sub-X Mounting Medium (Leica, cat. no. 3801741). On the day of the FISH experiment, we let the frozen slides reach −20 °C before transferring them directly to 1X PBS/4% paraformaldehyde (PFA) (VWR, cat. no. 10047105). We incubated the sections in PFA for 10 min at room temperature, followed by PFA quenching with 1X PBS/125 mM glycine and permeabilization in 1X PBS/0.5% Triton X-100 for 20 min at room temperature, followed by 5 min in 0.1 N HCl. To remove unspecific background fluorescence, we incubated the slides in 1X PBS/0.1 mg/mL RNase A (Sigma Aldrich, cat. no. R4875) at 37 °C for 1 h, followed by 1X PBS/0.5 mg/mL collagenase type 3 (Thermo Fisher Scientific, cat. no. 16111810) at 37 °C for 20 min. We performed successive dehydration steps in ethanol and let the slides air-dry. We transferred the slides to a buffer containing 2X SSC/50 mM sodium phosphate buffer (Thermo Fisher Scientific, cat. no. J60158.AP)/50% v/v formamide (Millipore, cat. no. S4117) and incubated them overnight in darkness at room temperature. On the following day, we covered the tissue sections with a pre-hybridization buffer containing 2X SSC/ 50 mM sodium phosphate buffer/50% formamide/5X Denhardt's solution (Invitrogen, cat. no. 750018)/1 mM EDTA (Sigma Aldrich, cat. no. AM9261)/100 µg/mL salmon sperm DNA (Thermo Fisher Scientific, cat. no. 15632011) and incubated them for 1 hour at 37 °C. We directly replaced this solution with hybridization buffer with the same composition plus 10% w/v dextran sulfate (Sigma Aldrich, cat. no. D8906) and the three FISH probes, each diluted at 0.05 nM per oligonucleotide. We covered the hybridization mix with an $18 \times 18$ mm$^2$ coverslip and sealed the sides of the coverslip with rubber cement (Fixogum, Triolab, cat. no. LK071A). We performed denaturation at 75 °C for 3 min followed by overnight incubation in a humidity chamber at 37 °C in darkness. After hybridization, we removed the coverslips inside a Petri dish pre-filled with 2X SSC/0.2% Tween-20, followed by washing off unbound probes twice in 0.2X SSC/0.2% Tween-20 at 60 °C for 7 min. We then pre-incubated the slides for 15 min in a buffer containing 2X SSC/25% v/v formamide before proceeding with secondary hybridization. We covered the samples with a hybridization buffer containing 2X SSC/25% v/v formamide/10% w/v dextran sulfate/1 mg/mL E. coli tRNA (Sigma Aldrich, Merck cat. no. 10109541001)/0.02% w/v bovine serum albumin (BSA, Thermo Fisher Scientific cat. no. AM2618) plus fluorescently conjugated detection oligonucleotides (purchased from Integrated DNA Technologies) diluted at 20 nM each. We incubated the slides covered with hybridization mix in a humidity chamber at 30 °C overnight. On the next day, we washed away unbound fluorescent oligos in 2X SSC/25% v/v formamide at 30 °C for 1 hour, followed by DNA staining in 2X SSC/25% v/v formamide/1 ng/µL Hoechst 33342 (Thermo Fisher Scientific, cat. no. 62249) for 1 hour at 30 °C. Finally, after washing the slides in 2X SSC, we mounted them in an anti-bleach imaging buffer containing 2X SSC/10 mM Tris-HCl/10 mM Trolox (Sigma Aldrich, cat. no. 238813)/37 ng/µL glucose oxidase (Sigma Aldrich, cat. no. G2133)/32 mM catalase (Sigma Aldrich, cat. no. C3515)/0.4% w/v glucose.

**Imaging.** We imaged all the samples on a custom-built Nikon T*i*-E Eclipse wide-field microscope equipped with an iXon Ultra 888 EMCCD camera (Andor Technology)[23]. We imaged the samples using a 25X/1.05 NA silicon oil-immersion objective (Nikon) for whole-section imaging and a 100X/1.45 NA oil-immersion objective (Nikon) for selected zoom-in regions. We acquired z-stacks spanning the entire slice thickness with a NIDAQ Piezo Z unit with an

interplane distance set at 450 and 300 nm for 25X and 100X magnification, respectively.

## Computational methods

**Sequencing data pre-processing.** After each sequencing run, we demultiplexed raw sequence reads to fastq files using the BaseSpace Sequence Hub cloud service of Illumina. In the case of scCUTseq, each fastq file typically contains 384 single cells. We further demultiplexed each fastq file using a custom Python script. In short, we extracted cell-specific barcodes and UMIs from each read at the specified nucleotide locations. Following this, we matched the barcodes to a list of predefined barcodes allowing for two mismatches if using a set of 384 adapters, or one mismatch if using a set of 96 adapters. We then re-wrote the reads to cell-specific fastq files with the barcode and UMI appended to the read name. For sequence reads that extended through adapters, we trimmed the adapters from cell- (or sample-) specific fastq files using fastp (version 0.20.1)[45]. We aligned the fastq files to the hg19 human reference genome using bwa-mem (version 0.7.17-r1188)[46] or to the dm6 *Drosophila melanogaster* reference in the case of S2 cells, and subsequently sorted and indexed using samtools (version 1.10)[47]. We moved barcodes and UMI tags from the bam file read header to the tags using a custom Python script. Finally, we deduplicated the reads based on the UMI tag and read position using umi-tools (version 1.1.1)[48]. In the case of WGS, we skipped the additional demultiplexing and trimming steps and deduplicated the reads using gatk MarkDuplicates (version 4.2.0.0)[49] instead. All the procedures described above are fully automated and streamlined using snakemake (version 5.30.1)[50].

**Copy number calling.** We counted reads in genomic bins of variable length (average length: 100 kb for bulk CUTseq; 250 kb for TK6 scCUTseq; 500 kb for all other scCUTseq and ACT datasets) based on mappability. Genomic bins that are located in low mappability regions are extended, while bins in high mappability regions are shortened. Following this, we filtered out blacklisted regions, including telomeric and centromeric regions, using a blacklist adapted from https://github.com/Boyle-Lab/Blacklist. The adapted blacklist is available at https://github.com/BiCroLab/scCUTseq/ under snakemake pipelines. We then normalized reads for library size and GC-content. Briefly, we calculated the ratio between read counts in each bin and the mean read counts across all bins and log-normalized the ratio. For each bin, we computed the GC-content and modeled a weighted linear regression between the GC-content and the normalized read counts using the LOWESS R function. We used this model to scale the read counts, normalizing for GC-content. Next, we either segmented the normalized read counts using the Circular Binary Segmentation module in the DNAcopy (version 1.66.0)[51] R package or, in the case of all breast and prostate single-cell data, joint segmentation using the multipcf function in the copynumber (version 1.29.0.9)[52] R package. Following this, we merged adjacent segments that were not significantly distinct using mergeLevels in the aCGH (version 1.78.0)[53] R package. In the case of single-cell data, we then inferred integer copy numbers using a grid search between different ploidy (ranging from 1.7 to 6, using 0.01 step sizes) and purity (being 1 for single-cell data) combinations and selecting the combination with the lowest error. We skipped this last step in the case of bulk sequencing data.

**Copy number quality control using a random forest classifier.** To exclude low-quality single-cell copy number profiles from our analyses, we trained a random forest classifier[54] based on 16 different copy number profile features (see Supplementary Data 2), using the profiles of 2304 single cells. In short, we manually annotated 2,304 scCUTseq profiles as high or low quality. We trained a random forest on 80% of these cells using the randomForest (version 4.6–14)[55] R package with

ntree = 500 and importance = TRUE ensuring class balance. To assess the performance of the random forest, we used the remaining 20% of the cells that were not used in the initial training as a validation set. Based on a receiver operating characteristic curve (ROC), we selected the most optimal threshold to classify single-cell copy number profiles.

**Cell cycle analysis using scAbsolute.** To assess whether some of copy number profiles discarded by our Random Forest classifier correspond to cycling cells in S phase, we applied the recently developed scAbsolute tool[20] to infer the cycling activity of 991 cells from four different scCUTseq libraries (MS101 and MS102 from prostate sample P2 and NZ235 and NZ236 from prostate sample P5, see Supplementary Data 7). We ran scAbsolute using the workflow described here: https://github.com/markowetzlab/scDNAseq-workflow. After obtaining the cycling activity for each cell in those libraries, we predicted, for each library separately, whether the cells were in S-phase using the predict_replicating() function in scAbsolute.

**Calculation of scCUTseq and ACT breadth of coverage and overdispersion.** We first downsampled single cells to 800 K reads. We then calculated the genome coverage using genomeCoverageBed from bedtools (version v2.25.0)[56] and the overdispersion by calculating the variance of read counts per bin normalized by the mean read counts per bin.

**Cell classification in tumor samples.** We classified cells in three different groups (diploid, pseudo-diploid, and aneuploid) based on their copy number profile. To this end, we calculated the percentage of the genome that was altered in each cell, meaning non-diploid copy numbers in autosomes and a non-diploid copy number state in chrX for female samples and non-haploid chrX for male samples. In the case of prostate samples, we then classified all the cells with no alterations as diploid; cells with less than 25% of the genome altered as pseudo-diploid; and cells with more than 25% of their genome altered as aneuploid. For breast cancer samples, we did not make a distinction between pseudo-diploid and aneuploid cells but classified all the cells harboring copy number alterations as (potential) tumor cells.

**Phylogenetic reconstruction and clone identification in tumor samples.** We constructed phylogenetic trees of pseudo-diploid prostate cells and of tumor breast cells using MEDICC2 (version 0.8.1)[21] with default parameters and total copy numbers of single cells as input. Following this, we used TreeCluster (version 1.0.3)[57] to cluster cells based on the Newick tree generated by MEDICC2. We used the 'max' clustering method, which clusters leaves (cells) so that the maximum distance between leaves within the same cluster is at most $t$, with $t$ equal to 3 and 4 for prostate samples P2 and P5, respectively, and 30 and 22 for breast cancer samples B1 and B2, respectively.

**UMAP on aneuploid cell SCNA profiles.** For aneuploid prostate cells, we first embedded the cells using UMAP with the following parameters: seed = 678, distance = 'manhattan', min_dist = 0 and n_neighbors = 6 (P2) and 3 (P5). Subsequently, we clustered the cells using the hdbscan function in the dbscan (version 1.1-8)[58] R package with minPts = 10.

**Pseudo-diploid subclone spatial distribution.** To quantify the spatial distribution of pseudo-diploid subclones in prostate samples, we calculated the Shannon entropy per clone as a proxy for how local or widespread clones are distributed. We normalized the number of cells from each clone based on the total number of cells that passed QC for each region and then calculated Shannon's entropy using the DescTools (version 0.99.49)[59] R package. An entropy close to 0 indicates highly localized distribution of a subclone while higher values indicate more widespread distributions.

**TCGA data analysis.** We downloaded segmented copy number profiles and masked somatic mutations of Prostate Adenocarcinoma samples (n = 499, TCGA, Firehose Legacy) from the cBioPortal[60] (https://www.cbioportal.org/) using the TCGAbiolinks (version 2.28.2)[61] R package. We classified patients in five different grade groups based on their Gleason score as following: 1) group 1: patients with 3 + 3 Gleason score; 2) group 2: patients with 3 + 4 Gleason score; 3) group 3: patients with 4 + 3 Gleason score; 4) group 4: patients with 4 + 4 Gleason score; and 5) group 5: patients with combined Gleason score 9 or 10. We then overlapped genomic regions that were either amplified, deleted or mutated with COSMIC and used ComplexHeatmap (version 2.16.0) to visualize the alterations. Finally, we used GISTIC2[62] (version 2.0.23) to look for genomic regions that are enriched for amplifications or deletions.

**DNA FISH image analysis.** We converted raw.nd2 image files to.tif format and performed 3D deconvolution using our Deconwolf software[24] using 100 iterations for all FISH channels and 50 iterations for the DNA channel. We reconstituted an overview of the entire tissue section by stitching the images acquired with 25x magnification, on which two pathologists independently manually annotated tumor and stroma regions using QuPath[63]. In parallel and blindly from the tumor annotation, we manually quantified FISH dots using ImageJ2[64] in 50 images acquired at 100x magnification and deconvolved using Deconwolf[24]. We then related the dot count of 'deletion' and 'upstream' probes to that of the 'downstream' probe (see Experimental Methods for a detailed description of the probes). Since the deletion is predicted to be monoallelic, we converted the relative proportion of deletion dots to the proportion of cells exhibiting the deletion in exactly one copy of chr13 using the following equation:

$$N_{cells\,with\,del.} = \frac{1 - N_{del.\,dots}}{0.5} \tag{1}$$

Finally, we related the proportion of cells harboring the deletion to the fraction of the 100x image covered by tumor lesions, according to the annotation performed by the pathologists on the stitched 25x magnification images.

### Reporting summary
Further information on research design is available in the Nature Portfolio Reporting Summary linked to this article.

## Data availability
The sequencing data (raw) for all the cell lines and breast cancer samples have been deposited in the European Nucleotide Archive (ENA) under accession code PRJEB71681. Supplementary Data 7 contains an overview of all sequencing runs and provides information whether the raw data are available on ENA, or not. The sequencing data (raw) for prostate, brain, and skeletal muscle samples cannot be publicly shared, either because the ethical permit for collecting the samples explicitly excluded it (prostate) or because explicit informed consent was lacking (brain and skeletal muscle). However, these data may be shared with individual researchers after sending a formal request to the corresponding author and only upon stipulation of a dedicated data transfer agreement, pending approval of the relevant ethical review board. An initial response to a request may be expected from the corresponding author within two weeks. All sequencing data sets have been aligned to the hg19 reference genome. Processed sequencing data for all samples including prostate, brain, and skeletal muscle are available on Figshare[65]. Tabulated data to reproduce all the plots in the main Figures and Supplementary Figs. are available as a Source Data file. The TCGA data that we used here can be downloaded from https://www.cbioportal.org/study/cnSegments?id=prad_tcga. Source data are provided with this paper.

## Code availability
All the custom code used for processing the sequencing data generated by scCUTseq is available on GitHub (https://github.com/BiCroLab/scCUTseq and https://bicrolab.github.io/scCUTseq/) and zenodo[66].

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

## Acknowledgements

We thank Britta A.M. Bouwman (Bienko-Crosetto Lab) for critically reading the manuscript and helping with editorial revisions; Jinxin Chen

(Bienko-Crosetto Lab) for uploading the raw sequencing data to the European Nucleotide Archive; Grazyna Lietzau (Stockholm Southern Hospital) for help with sectioning and H&E staining of prostate tissues; Maxime Tarabichi (Université Libre de Bruxelles) for help with bioinformatic analyzes; Cecilia Söderberg-Nauclér (Karolinska Institutet) for providing access to a microscope slide scanner to digitalize H&E-stained tissue sections; Vera Minneker (Roukos lab) for generating the TK6-Cas9 expressing cell line; and Daniela Cantarella (Sequencing Facility at Candiolo Cancer Institute) for preparing RNA-seq libraries. We acknowledge the Biomedicum Flow cytometry Core facility (Karolinska Institutet), supported by KI/SLL, for providing cell sorting services; the National Genomics Infrastructure in Stockholm funded by Science for Life Laboratory, the Knut and Alice Wallenberg Foundation and the Swedish Research Council for sequencing on the NovaSeq 6000; the SNIC/Uppsala Multidisciplinary Center for Advanced Computational Science for assistance with massively parallel sequencing and access to the UPPMAX computational infrastructure for processing NovaSeq data; the IMB Microscopy Core Facility for providing access to the Opera Phenix microscope (funded by the Deutsche Forschungsgemeinschaft, grant. no. INST 247/845-1 FUGG) for detecting and assessing the frequency of the CRISPR-Cas9 induced deletion on chr11 in TK6 cells. This work was supported by grants from the National Natural Science Foundation of China (grant no. 81972475, 82373267) and the Mount Taishan Scholar Young Expert (NO.tsqn202306345) to N.Z.; by a scholarship from the Marie Sklodowska-Curie Innovative Training Networks (ITN) (H2020-MSCA-ITN-2018, grant no: 812829 "aDDRess") to L.H.; by a scholarship from MIUR - Dipartimenti di Eccellenza 2018–2022 (Project No. D15D18000410001) to E.B.; by grants from the Swedish Research Council (grant no. 2013-07618) and the Swedish Cancer Society (grant no. 22 2358 Pj) to J.F.; by grants from the Swedish Research Council (grant no. 2015-00162) and the Swedish Cancer Society (grant no. CAN2018/0658) to T.H.; by grants from the Deutsche Forschungsgemeinschaft (DFG, grant no. 393547839–SFB1361, 402733153–SPP2202 and 455784893) to V.R.; by a grant from Fondazione AIRC per la Ricerca sul Cancro (grant no. 22850 under IG 2019) to C.M.; by grants from the Swedish Cancer Society (grant no. 19 0130 Pj 03 H) and from the Ragnar Söderberg Foundation (Fellows in Medicine 2016) to M.B.; by grants from the Swedish Cancer Society (grants no. CAN 2018/728 and 21 1785 Pj), the Strategic Research Programme in Cancer (StratCan) and Cancer Research KI at Karolinska Institutet (grant. no. 2201), the Fondazione Piemontese per la Ricerca sul Cancro (FPRC 5xmille 2018 Ministero Salute, Programma ADVANCE -SINCE), and the Italian Ministry of Health (Conto Capitale E.F. 2019-2020 and Ricerca Corrente 2024) to N.C.; and by a joint grant from the Swedish Foundation for Strategic Research (SSF, grant no. BD15_0095) to N.C. and T.H.

## Author contributions

Project conceptualization: N.C. scCUTseq development and application: N.Z. and M.S. Validation by ACT: C.D. Validation by DNA FISH: Q.V. CRISPR experiments: G.L. and V.R. scCUTseq on brain and skeletal muscle samples: M.R., R.B., D.G. RNA-seq: E.B. and S.B. Targeted sequencing: E.B. Breast cancer samples and annotations: C.M. and A.S. Brain and skeletal muscle samples: M.R., K.A., H.D. and J.F. Prostate cancer samples: F.T. and N.S. Prostate samples pathological annotation: P.S., B.H., and W.W. Clinical knowledge support: Q.Y. and N.Z. Data analysis and visualization: L.H. Random Forest classifier: L.H. and S.O. Funding acquisition: N.C., M.B., T.H., J.F., V.R., S.G., A.S. and C.M. Supervision and project coordination: N.C. Figures: N.C. and M.B. Writing: N.C. with input from all the authors.

## Funding

## Competing interests

The authors declare no competing interests.
