## [Peer Review File · Nature Communications]

REVIEWER COMMENTS

Reviewer #1 (Remarks to the Author):

I would like to commend the authors on a remarkable amount of new data that has substantially improved the paper. I have only a few relatively minor points to suggest, and retain the numbering/lettering scheme from the prior review for clarity.

1. scCUTseq

b) I think the combination of the new data and these explanations are fairly convincing. However I would like the authors to ensure that something approximating this reply is added to the manuscript in an appropriate place to make this rationale available in some detail to readers, as I think it's a natural question that will arise in reading this work.

d) I note that it would be useful to quantitatively assess the range of resolutions achieved across different samples, but I appreciate the biological focus of the revised manuscript and agree that would be orthogonal to the current work.

h) Not having data availability to the raw sequencing data notably diminishes the value of this study, and may not be compliant with NPG policies (but of course that is an editorial consideration, not a review one!)

2. Breast Cancer

c) The updated trees (and tree generation strategy) are significantly improved throughout. Nice work!

3. Prostate Cancer

a) Ahhh, this is very interesting, please consider moving Supplemental Figure 10 into the main text, it is a very important result.

f) I am rather skeptical of the highlighting of somatic deletion of BRCA1, RB1 and FOXO1/3, which are generally not considered to be amongst the most critical drivers of early prostate cancer evolution. Rather, it would have been anticipated to see deletions of NKX3-1, PTEN and to a lesser extent BRCA2 and TP53. I would like to see some thoughtful discussion of the concordance/discordance of the results here with TCGA/ICGC (both PRAD-CA and PRAD-UK) results. Discussion and citation of relevant literature only, not suggesting significant new experimental work here.

h) Fascinating results, please include detailed analysis of the TSO500 results in a supplemental figure as this will be quite influential (unless it's already there and I missed it, in which case my apologies)

Reviewer #4 (Remarks to the Author): Expert in single-cell DNA sequencing, cancer genomics and evolution; replaces Reviewer #2

I have reviewed the revised manuscript which was transferred from another Nature journal after revisions in response to reviewer comments.

Overall the authors have carefully and sincerely attempted to address the major comments of the reviewers and I believe the most important criticisms appear to have been addressed.

The previous reviewers raised many points I would have agreed with, and some that, in my view, were misplaced - for example, the requirement to infer phylogeny from the pseudo-diploid cells ignores the possibility that single cell sequencing of tissues may reveal events distributed over many individual cells, the products of single mitoses, which all differ in exact genotype. Such a tree would be all leaves and not very informative. Perhaps more relevant would be an analysis of the distribution and size of copy number events and breakpoints in relation to chromosomal location and the most commonly reported CNA events in bulk sequenced prostate cancers.

After reading the manuscript and responses I note some minor points which should be considered to help the clarity/technical documentation of the work before publication:

A minor technical point that I could not see raised in any of the reviews relates to duplicate cells and s-phase classification of single cell genomes. All single cell methods have some duplicate cell collision rate, this should be reported for the SCUT method so that the possibility of collision genotypes can be assessed in the data, especially with respect to the highly aneuploid cells reported as "monsters".

The second feature, s-phase state identification, is highly pertinent to copy number calling as partially replicated cells will exhibit genomes that appear with deletions. Its likely the random forest classifier trained on "low quality" cells identified by inspection may gate out many or even the majority of replicating cells, as high dispersion and other features used in their random forrest are also features of s-phase cells. Nevertheless I could not see this addressed specifically in the manuscript or previous reviewer comments in a satisfactory way. Alternative probabilistic approaches to this issue have recently been released on bioarxiv (eg PERT, scAbsolute). I am not suggesting the s-phase fraction needs to be explicitly enumerated, but at least the authors should document that the random forrest classifier as used efficiently tags/removes the majority of such cells from downstream analysis.

A further issue that influences copy segment calling is GC mappability correction and assignment of ploidy. The authors should better document how these issues are handled and their effects in their CNA pipeline.

The comparisons of costs and discussion thereof are not rigorous and do not compare all methods. For example the costs of labor associated with the different processes is not included, the effect of the methods on junk sequencing such as duplicate rates, also has an effect on costs (ie the sequencing efficiency is not equal necessarily between methods), the consequences of highly scaled down libraries in smaller than 384 well plates which is described for DLP+, not compared. I don't believe these comparisons are rigorous or pertinent to the value of the manuscript and could be safely left out of the discussion.

I agree with the previous reviewers that the term "monster cells" while entertaining is not especially informative. Monster could be mean anything without context. They are highly aneuploid cells that differ in degree from low aneuploidy cells. I would encourage the authors to consider which has the greatest transparency for non-expert readership who many not have seen the one paper cited that mentions "monsters".

The manuscript could still be clearer on what is the consensus pseudobulk CNA genotype of the two prostate cancers studied vs the pileup consensus of pseudo-diploid cells looks like - and how this differs or overlaps with TCGA WGS profiles - the oncogene mapping is helpful, but perhaps comparative Manhattan plots or some similar approach would reveal it more clearly.

Reviewer #5 (Remarks to the Author): Expert in spatial and single-cell genomics, functional genomics, and cancer genomics; replaces Reviewer #3

This manuscript describes the highly diverse clonal structure in prostate cancers. To identify this diversity, the authors developed and utilized the single cell analysis method, named scCUT. After the technical evaluation, they applied the developed method to the analysis of clinical specimens. Overall, I consider this manuscript should convey interesting insights. However, I have technical concerns about scCUT sequencing as follows.

1. The presented whole genome sequencing analysis is shallow. Even though the authors present the results of regional WGSs (Supplementary Figure 10), most of the SCNAs still remain elusive. If they are genuine, they would be detected by the increasing sequencing depth. More directly, especially for short deletions, there should be a reasonable chance that their junctions could be identified by “software-clipped (or bridging) sequences” at least a relevant part of them. For such a bulk analysis, the non-PCR library would be more plausible. After all, for the bulk analyses, relatively wide regions should be (and actually have been) dissected and used for the DNA extraction. If so, even long read sequencing can be contemplated, especially for the validation of SCNAs occurring in repetitive or other difficult regions.

2. As for the bulk WGS, even more quantitative analyses could be done as to whether the clonal composition could be de-convolved to explain the observed frequencies of SCNAs by scCUT. Conversely, if the junctions of the given frequency could not be supported at a given sequencing depth, the results should be considered with that caveat. I hope, even if the possible error frequencies should be taken into consideration, the remaining population should draw essentially the same conclusions.

3. Phylogenetic and clonal structure analyses should be further deepened. As appearing in Supplementary Fig, 13, each region (Column x Row) includes a large number of sub-clones at the complex compositions and their compositions seemed unrelated even in their neighboring regions (assuming the near column and row IDs should represent physical neighborhood; please include exact location of each region on the HE images, perhaps using the consecutive sections). Intuitively, I wonder how the clonal evolution (or random occurrence and selection for each clone) was possible to form this structure.

Point-by-point response to the Reviewers' comments

We would like to thank once again the original Reviewers for their time as well as the newly included Referees for assessing our revised manuscript and providing further inputs. In this second round of revisions, we have strived to address all the remaining Reviewers' remarks by performing additional analyses and re-organizing several figures/text parts, aiming at further strengthening our manuscript and making it suitable for publication in *Nature Communications*. Specifically:

- Following the suggestion of Reviewer #1, we have moved the previous **Supplementary Fig. 10**, comparing scCUTseq with bulk WGS, to **Figure 1** (see **new Fig. 1o, p**), to better showcase the fact that subclonal deletions detected by scCUTseq across multiple cells are also recapitulated in bulk WGS, strongly indicating that these are not artefacts of scCUTseq.
- As suggested by the same Reviewer, we have further expanded our analysis of the TSO500 data and now provide OncoPrint plots summarizing the type and spatial distribution of mutations across the tissue regions analyzed (see **new Supplementary Fig. 23e and f**).
- Following the suggestion of Reviewer #4, we have implemented the scAbsolute tool to computationally pinpoint cells that are likely in S-phase. We selected two regions in each of the two prostate samples (P2 and P5) that we thoroughly profiled by scCUTseq (991 cells in total) and used scAbsolute to infer the so-called cycling activity threshold to detect S-phase cells. As shown in the **new Supplementary Fig. 10**, virtually all cells annotated as replicating by scAbsolute had been filtered out by our random forest classifier, indicating that these cells are not technical failures but rather cells in S-phase.
- Following the remarks of Reviewer #5, we have further analyzed our WGS data to assess whether we could detect more instances of the deletion events captured by scCUTseq. Unfortunately, since we could only perform shallow WGS on these samples (compatibly with the available project budget), we were not able to extract more information from the existing data. We also note that performing deeper WGS or even long-read sequencing, as proposed by the Reviewer, in our view would go beyond the scope of this work, since we have amply demonstrated the reliability of scCUTseq and do not think that further validation by bulk approaches is needed.

In sum, we thank the Reviewers' once again for their valuable comments and suggestions, which helped us further strengthen our manuscript in this second round of revisions, and hope that they are now supportive of publication of our work in *Nature Communications*.

Reviewer #1

I would like to commend the authors on a remarkable amount of new data that has substantially improved the paper. I have only a few relatively minor points to suggest, and retain the numbering/lettering scheme from the prior review for clarity.

We would like to express our gratitude to the Reviewer for their valuable time and suggestions and for appreciating our continued effort to strengthen our manuscript. We hope that the Reviewer will be satisfied by our additional revisions and find our manuscript now suitable for publication in *Nature Communications*.

1. scCUTseq

b) I think the combination of the new data and these explanations are fairly convincing. However I would like the authors to ensure that something approximating this reply is added to the manuscript in an appropriate place to make this rationale available in some detail to readers, as I think it's a natural question that will arise in reading this work.

We are pleased that the Reviewer found our new data and explanations convincing. We have tried to synthesize our previous reply to this comment in this newly added paragraph in the Discussion section:

<< One inherent limitation of scCUTseq is the sparse breadth of genome coverage that can be achieved (typically, less than 2%) when single restriction enzymes are used, as in this study. This limits the theoretical maximum genomic resolution that can be achieved (as we previously extensively discussed for bulk CUTseq¹⁶) and precludes the possibility of determining B-allele frequencies and hence obtaining phased copy number profiles. However, as we have shown in this study, scCUTseq detects very similar copy number profiles and cell populations as tagmentation-based ACT, indicating that the use of restriction enzymes is fully compatible with single-cell SCNA profiling in tumors and other tissues. Of note, another scDNA-seq method named Karyo-Seq also leverages restriction enzymes and was used to profile SCNAs in patient-derived tumor organoids^{40,41}. >>

d) I note that it would be useful to quantitatively assess the range of resolutions achieved across different samples, but I appreciate the biological focus of the revised manuscript and agree that would be orthogonal to the current work.

We agree that it would be valuable to quantitatively assess and report the range of resolutions that can be achieved with scCUTseq. However, we appreciate that the Reviewer agrees that this should be addressed in a more technically focused follow-up study.

h) Not having data availability to the raw sequencing data notably diminishes the value of this study, and may not be compliant with NPG policies (but of course that is an editorial consideration, not a review one!)

We thank the Reviewer for raising this important issue, which we have also discussed with the Editor. We have now created a dedicated project in ENA (<https://www.ebi.ac.uk/ena/browser/view/PRJEB71681>) and are currently in the

process of uploading the raw sequencing data obtained from all the cell lines and breast cancer samples described in this study, as described in the Data availability statement. However, due to a technical failure in one of our servers, we are experiencing some delays in the upload process, which we hope to resolve as soon as possible.

Unfortunately, we are unable to upload to the same ENA project also the raw sequencing data files obtained from the prostate, brain, and skeletal muscle samples described in this study. In the case of prostate samples, the reason is that the ethical permit for collecting these samples explicitly excluded sharing of raw sequencing data. In the case of brain and skeletal muscle samples, the ethical permit for collecting these (autopsy) samples dates back to 2013 and the donors did not provide explicit informed consent to publically share any raw sequencing data eventually produced from their samples.

Nonetheless, since we are big supporters of open science, we have thought of the following compromise solution, which we have now included in the Data Availability statement:

<< Therefore, these data may be shared with individual researchers requesting them only upon stipulation of a dedicated data transfer agreement, pending approval of the relevant ethical review board. >>

We acknowledge that this legalistic solution might discourage many groups from requesting our data, but this is unfortunately the only option that we can pursue, as we have also communicated to the Editor. We do note, however, that the Readers should be able to reproduce all the analyses and plots presented in our manuscript, using the fully anonymized pre-processed data and scripts that we provide at <https://bicrolab.github.io/scCUTseq/>.

2. Breast Cancer

c) The updated trees (and tree generation strategy) are significantly improved throughout. Nice work!

We are grateful to the Reviewer for their appreciation and nice words.

3. Prostate Cancer

a) Ahhh, this is very interesting, please consider moving Supplemental Figure 10 into the main text, it is a very important result.

We thank the Reviewer for suggesting this. Accordingly, we have now moved the previous **Supplementary Fig. 10** to **Figure 1** (see **new Fig. 1o, p**) and adjusted the text accordingly.

f) I am rather skeptical of the highlighting of somatic deletion of BRCA1, RB1 and FOXO1/3, which are generally not considered to be amongst the most critical drivers of early prostate cancer evolution. Rather, it would have been anticipated to see deletions of NKX3-1, PTEN and to a lesser extent BRCA2 and TP53. I would like to see some thoughtful discussion of the concordance/discordance of the results here

with TCGA/ICGC (both PRAD-CA and PRAD-UK) results. Discussion and citation of relevant literature only, not suggesting significant new experimental work here.

We thank the Reviewer for raising this important point. Indeed, NKX3-1, PTEN, BRCA2, and TP53 are commonly lost in prostate cancer. We did detect BRCA2 deletions in one of the two (P2) prostate samples that we thoroughly profiled by scCUTseq (in fact, BRCA2 was the most frequently deleted gene in the TRR/FER-specific pseudo-diploid subclones found in this sample, see **Fig. 4a**), while we did not detect NKX3-1, PTEN or TP53 deletions (see **Supplementary Table 6**). We do note, however, that while these deletions are relatively frequent in TCGA prostate cancers, we only profiled a very small subset of patients and therefore it should not be surprising that the spectrum of genes that we found altered does not exactly match the one from large cohort studies such as PRAD-CA and PRAD-UK. We have now added this sentence in the paragraph in the Results section describing cancer genes deleted in pseudo-diploid cells:

<< We did not find deletions affecting NKX3-1, PTEN and TP53 genes, which are frequently deleted in TCGA PRADs. However, this is likely due to the very limited sample size of our proof-of-concept study. >>

h) Fascinating results, please include detailed analysis of the TSO500 results in a supplemental figure as this will be quite influential (unless it's already there and I missed it, in which case my apologies)

We are glad that the Reviewer appreciated our expansion of the TSO500 dataset and thank them for their suggestion to better portray this effort. Accordingly, in the **new Supplementary Fig. 23e and f** we now present two heatmaps showing all the detected SNVs and indels in each of the tissue regions profiled by TSO500 and scCUTseq in samples P2 and P5.

Reviewer #4

I have reviewed the revised manuscript which was transferred from another Nature journal after revisions in response to reviewer comments.

Overall the authors have carefully and sincerely attempted to address the major comments of the reviewers and I believe the most important criticisms appear to have been addressed.

We would like to sincerely thank the Reviewer for their time and willingness to assess our response to the previous round of Reviewers' comments at *Nature*. We are pleased to see that the Reviewer appreciates our efforts in revising our original manuscript and thank them for the additional suggestions, which we believe have helped us further strengthen our work. We hope that, following this second round of revisions, the Reviewer will be supportive of publication of our work in *Nature Communications*.

The previous reviewers raised many points I would have agreed with, and some that, in my view, were misplaced - for example, the requirement to infer phylogeny from the pseudo-diploid cells ignores the possibility that single cell sequencing of tissues may reveal events distributed over many individual cells, the products of single mitoses, which all differ in exact genotype. Such a tree would be all leaves and not very informative. Perhaps more relevant would be an analysis of the distribution and size of copy number events and breakpoints in relation to chromosomal location and the most commonly reported CNA events in bulk sequenced prostate cancers.

We are thankful to the Reviewer for highlighting this important consideration. We agree that additional analyses of breakpoints and size of CNA events would be of great interest. However, it is crucial to note that our current copy number calling is performed at 500 kb resolution. In contrast, the publically available copy number profiles from TCGA were generated using SNP arrays, which have a much higher resolution compared to scCUTseq. This discrepancy results in a considerable difference in the range of detectable CNA sizes and breakpoint detection accuracy, making a direct comparison of breakpoints and CNA sizes detected by scCUTseq with TCGA challenging. Moreover, we would like to emphasize that, while we have generated a rich dataset from hundreds of single cells from spatially annotated tissue regions, our data only come from two patients (with limited extra data from four additional patient samples for technical validation purposes). Hence, a thorough comparison of CNA events between our dataset and TCGA, in our view, would not be statistically sound and unlikely lead to conclusions that could be generalized.

We would like to emphasize that the primary motivation for us to conduct this project was to demonstrate that the scCUTseq method, which we developed, can be applied in a multi-region single-cell sequencing approach, to gain insights into spatial genetic heterogeneity, in a model of early tumor evolution. Follow-up applications of scCUTseq or other scDNA-seq techniques (such as ACT or DLP+) to a much larger set of samples will therefore be needed to extend our observations and make meaningful comparisons with publically available CNA datasets such as TCGA.

After reading the manuscript and responses I note some minor points which should be considered to help the clarity/technical documentation of the work before publication:

A minor technical point that I could not see raised in any of the reviews relates to duplicate cells and s-phase classification of single cell genomes. All single cell methods have some duplicate cell collision rate, this should be reported for the SCUT method so that the possibility of collision genotypes can be assessed in the data, especially with respect to the highly aneuploid cells reported as "monsters".

We thank the Reviewer for raising this important point. Since we use FACS to sort single cells in plates, we cannot exclude that some wells will end up containing two cells instead of one, even though we try to minimize this risk by setting very stringent sorting parameters. While setting scCUTseq up back in 2020, we tested this possibility by sorting cells in glass-bottom 96-well plates and indeed could detect cell pairs in few wells. However, even in the unlikely case of one diploid cell and one cell with one or more deletions ending up in the same well, this would not result in the aneuploid copy number profiles that we detect in 'monster' cells. Thus, we are confident that the majority of 'monster' cells that we identified represent truly aneuploid cells, likely originating from mitotic errors.

The second feature, s-phase state identification, is highly pertinent to copy number calling as partially replicated cells will exhibit genomes that appear with deletions. Its likely the random forest classifier trained on "low quality" cells identified by inspection may gate out many or even the majority of replicating cells, as high dispersion and other features used in their random forrest are also features of s-phase cells. Nevertheless I could not see this addressed specifically in the manuscript or previous reviewer comments in a satisfactory way. Alternative probabilistic approaches to this issue have recently been released on bioarxiv (eg PERT, scAbsolute). I am not suggesting the s-phase fraction needs to be explicitly enumerated, but at least the authors should document that the random forrest classifier as used efficiently tags/removes the majority of such cells from downstream analysis.

We are very grateful to the Reviewer for raising this important issue and for bringing these tools to our attention. Following the Reviewer's suggestion, we have now applied scAbsolute to computationally pinpoint cells that are likely in S-phase. We selected two regions in each of the two prostate samples (P2 and P5) that we thoroughly profiled by scCUTseq (991 cells in total) and used scAbsolute to infer the so-called cycling activity threshold to detect S-phase cells. Next, we compared whether the putative S-phase cells identified had been filtered out by our random forest classifier. Indeed, virtually all cells annotated as in S-phase by scAbsolute had been filtered out by our classifier, indicating that these cells are not technical failures but rather replicating cells. We now present the results of this analysis in the **new Supplementary Fig. 10**. However, we have not extended this analysis to the whole dataset, as this would require a considerable amount of time and further delay publication of our work. We hope that the Reviewer will find this acceptable.

A further issue that influences copy segment calling is GC mappability correction and assignment of ploidy. The authors should better document how these issues are handled and their effects in their CNA pipeline.

Thanks for touching upon this point. As we have described in the Methods section (Computational methods >> Copy number calling), we have accounted for differences in mappability across genomic regions by using a variable width binning approach. Using this approach, genomic regions with low mappability will result in larger bin sizes, ensuring that each bin has approximately the same number of reads (in the absence of copy number events). Furthermore, we normalized the read counts for GC-content using LOWESS smoothing. Finally, to infer the integer copy number, since we are dealing with single cells we set the purity to 1 and test a range of different ploidy values (ranging from 1.7 to 6, using 0.01 steps), we then select the ploidy value that results in the lowest error in final integer copy number calls. We now tried to improve the description of these steps in the corresponding Methods section, hoping that the Reviewer will find this part clearer.

The comparisons of costs and discussion thereof are not rigorous and do not compare all methods. For example the costs of labor associated with the different processes is not included, the effect of the methods on junk sequencing such as duplicate rates, also has an effect on costs (ie the sequencing efficiency is not equal necessarily between methods), the consequences of highly scaled down libraries in smaller than 384 well plates which is described for DLP+, not compared. I don't believe these comparisons are rigorous or pertinent to the value of the manuscript and could be safely left out of the discussion.

We agree with the Reviewer that our cost analysis is incomplete as it does not consider all the scDNA-seq approaches that have emerged since we developed scCUTseq, including microfluidic-based methods such as DLP+. Accordingly, we have now removed the Cost Analysis and related Supplementary Table 10 from the Supplementary Information, only stating that scCUTseq is a cost-effective method in the Discussion section.

I agree with the previous reviewers that the term "monster cells" while entertaining is not especially informative. Monster could be mean anything without context. They are highly aneuploid cells that differ in degree from low aneuploidy cells. I would encourage the authors to consider which has the greatest transparency for non-expert readership who many not have seen the one paper cited that mentions "monsters".

We acknowledge that the term 'monster cells' is divisive as more than one Reviewer has argued against its use. We have therefore decided to remove this term from the manuscript and figures and use the term 'aneuploid cells' instead. However, we have kept the reference to the paper previously published in *Nature Genetics* (PMID: 34211178) that introduced this term. We hope that the Reviewer will find this solution acceptable.

The manuscript could still be clearer on what is the consensus pseudobulk CNA genotype of the two prostate cancers studied vs the pileup consensus of pseudo-diploid cells looks like - and how this differs or overlaps with TCGA WGS profiles - the oncogene mapping is helpful, but perhaps comparative Manhattan plots or some similar approach would reveal it more clearly.

We appreciate the valuable suggestion from the Reviewer. Accordingly, we have now performed an additional analysis to explore whether the most frequently altered genomic regions in the two prostate samples that we thoroughly profiled by scCUTseq (P2 and P5) overlap with frequently altered regions identified in TCGA PRAD data (see **Fig. 4 j**). As it can be seen in the Manhattan plots below (suggested by the Reviewer), P2 clearly exhibits a high frequency of deletions in chromosome 6 and 13, resembling commonly deleted regions detected in TCGA PRAD. In contrast, P5 does not show a clear overlap with TCGA PRAD in terms of frequently deleted regions.

As outlined in our previous response to the Reviewer's comment on comparison of CNA sizes and breakpoints between our dataset and TCGA, it is crucial to interpret these findings with a grain of salt, considering our very limited sample size. In fact, even the most common alterations reported in TCGA PRAD will not be universally present in all prostate tumors sequenced, even if they come from patients of similar ethnicity. Given that the primary focus of our study was not to directly compare alterations and affected genes to publically available prostate cancer data, we would prefer to show this additional analysis only to the Reviewer and not include it in the revised manuscript. However, if the Reviewer thinks these additional results should be available to future readers of our paper, we will be happy to include them in one additional supplementary figure.

Reviewer #5

This manuscript describes the highly diverse clonal structure in prostate cancers. To identify this diversity, the authors developed and utilized the single cell analysis method, named scCUT. After the technical evaluation, they applied the developed method to the analysis of clinical specimens. Overall, I consider this manuscript should convey interesting insights. However, I have technical concerns about scCUT sequencing as follows.

1. The presented whole genome sequencing analysis is shallow. Even though the authors present the results of regional WGSs (Supplementary Figure 10), most of the SCNAs still remain elusive. If they are genuine, they would be detected by the increasing sequencing depth. More directly, especially for short deletions, there should be a reasonable chance that their junctions could be identified by “software-clipped (or bridging) sequences” at least a relevant part of them. For such a bulk analysis, the non-PCR library would be more plausible. After all, for the bulk analyses, relatively wide regions should be (and actually have been) dissected and used for the DNA extraction. If so, even long read sequencing can be contemplated, especially for the validation of SCNAs occurring in repetitive or other difficult regions.

We thank the Reviewer for assessing our revised manuscript and suggesting additional actions that we could take to further validate the deletions detected by scCUTseq. While in principle we agree that deep WGS and long-read sequencing would be greatly beneficial, allowing us to accurately identify deletion breakpoints and hence further validating our single-cell data, such experiments fall well out of the scope and feasibility of this work, mainly because the budget that we were granted to conduct this study did not account for multi-region deep WGS and long-read sequencing. Importantly, in our revised manuscript we now provide multiple levels of validation of the deletions detected by scCUTseq:

1. Accurate and sensitive detection of a single deletion induced by CRISPR-Cas (see **Supplementary Fig. 2**).
2. Thorough comparison with a WGA-free scDNA-seq method (ACT), recapitulating all the major findings by scCUTseq (see **Fig. 3a and Supplementary Fig. 3**).
3. Detection of several subclonal deletions previously identified by scCUTseq using shallow WGS (see **new Fig. 1o, p**). Notably, using shallow WGS we are able to detect a log₂ratio drop suggesting the presence of a deletion even for genomic regions detected as deleted by scCUTseq in as little as two cells (see, for example, the log₂ratio drop on chr8 p-arm in **new Fig. 1o**).
4. Validation of one subclonal deletion detected by both scCUTseq and ACT using an orthogonal method, DNA FISH (see **Fig. 3**).

We believe this cumulative evidence convincingly demonstrates that the deletions marking the pseudo-diploid cells identified by scCUTseq are real and not an artefact, arguing against the need of further validation experiments. We also note that the deletions that we have only detected in a single cell likely represent unique events that would be hardly detectable using standard 40-60X WGS. Deeper sequencing might uncover such events, but at a cost that could not be covered with the budget that we were granted to conduct this study. We therefore sincerely hope

that the Reviewer will agree that deeper WGS and long-read sequencing should be considered for future follow-up studies, ideally conducted on a larger patient cohort.

2. As for the bulk WGS, even more quantitative analyses could be done as to whether the clonal composition could be de-convolved to explain the observed frequencies of SCNAs by scCUT. Conversely, if the junctions of the given frequency could not be supported at a given sequencing depth, the results should be considered with that caveat. I hope, even if the possible error frequencies should be taken into consideration, the remaining population should draw essentially the same conclusions.

We agree with the Reviewer that our WGS analysis could be improved by including BAF and purity estimates, which would enable us to infer the subclonal fraction of specific copy number events and compare this to frequencies detected by scCUTseq. Unfortunately, when we tried to obtain this information using our shallow WGS data, we were not able to get reliable BAF estimates due to the sequencing depth being insufficient. Since, as already explained above, deeper sequencing of these libraries would not be feasible under the budget that we were granted to conduct this project, we hope that the Reviewer is open to consider our current WGS analysis, combined with all the additional levels of validation listed above, as sufficient to demonstrate that the CNAs we detected are true events.

3. Phylogenetic and clonal structure analyses should be further deepened. As appearing in Supplementary Fig, 13, each region (Column x Row) includes a large number of sub-clones at the complex compositions and their compositions seemed unrelated even in their neighboring regions (assuming the near column and row IDs should represent physical neighborhood; please include exact location of each region on the HE images, perhaps using the consecutive sections). Intuitively, I wonder how the clonal evolution (or random occurrence and selection for each clone) was possible to form this structure.

We thank the Reviewer for raising this important point. Indeed, pie charts in the previous **Supplementary Fig. 13** (now **Supplementary Fig. 15**) are arranged based on the position of the corresponding tissue block (region) in the prostate midsection, as outlined in **Fig. 1c, d**. To make this figure clearer, we now show the same tissue maps as in Fig. 1c, d also in Supplementary Fig. 15 and have added a colorbar indicating the corresponding region type (tumor-rich, focally enriched or normal) based on morphological classification. Furthermore, we have added back two supplementary figures (**new Supplementary Fig. 5 and 6**), which we originally displayed in the first submission to *Nature* and later removed during the first revision, showing the hematoxylin-eosin stained top and bottom sections for tumor-rich (TRR) and focally enriched (FER) regions, with tumor regions identified by two independent pathologists highlighted in yellow in each section.

Regarding the large variability in the number of subclones identified, even between adjacent regions, we concur with the Reviewer that this is rather unexpected and intriguing. We hypothesize that this phenomenon stems from widespread ongoing chromosomal instability across the entire prostate, as highlighted by the detection of numerous aneuploid cells even in morphologically normal tissue sections. This chromosomal instability likely fosters the emergence of multiple distinct SCNA

subclonal events across the entire prostate, explaining the high heterogeneity observed and absence of spatial correlation between different subclones, although some of them are clearly enriched in multiple adjacent regions (see, for example, clone c1 in sample P5 in the **new Supplementary Fig. 15**).

REVIEWERS' COMMENTS

Reviewer #1 (Remarks to the Author):

Outstanding revisions on both rounds here, and I am very strongly in favour of acceptance as-is.

Reviewer #4 (Remarks to the Author):

The authors have addressed the most important comments that I raised and I have no new substantive comments to make.

Apropos the Manhattan plots of the CNA landscape, I understand the point the authors are making but it will be easier for future readers if these were in fact incorporated in the supplemental material, ideally with a cosine distance or similar metric to TCGA PRAD CNA landscapes for each tumour landscape, to give some quantitative meaning to the degree similarity.

Reviewer #5 (Remarks to the Author):

First of all, I appreciate the authors' substantial efforts to improve this manuscript. Owing to their newly generated results from the extensive analyses. I consider this manuscript should have been very much improved. Even though I still have a remaining concern regarding the shallow depth of the WGS presented here and the technical limits of their scCUTseq, I understand that this part should wait for their future work. I sincerely hope the authors further continue the spatial analysis of the cancer cell diversities and will have a change to look back, again, on the results appearing in this particular manuscript.

Point-by-point response to the Reviewers' comments

Reviewer #1

Outstanding revisions on both rounds here, and I am very strongly in favour of acceptance as-is.

We are grateful to the Reviewer for their kind words and for being supportive of publication of our manuscript in *Nature Communications*.

Reviewer #4

The authors have addressed the most important comments that I raised and I have no new substantive comments to make.

Apropos the Manhattan plots of the CNA landscape, I understand the point the authors are making but it will be easier for future readers if these were in fact incorporated in the supplemental material, ideally with a cosine distance or similar metric to TCGA PRAD CNA landscapes for each tumour landscape, to give some quantitative meaning to the degree similarity.

We thank the Reviewer for their time and insightful comments and are glad that the Reviewer are satisfied by our revisions.

Regarding the Manhattan plots, we have followed the Reviewer's suggestion and now include these plots in the **new Supplementary Fig. 24** (previously Supplementary Fig. 23) and cite them in the main text.

Reviewer #5

First of all, I appreciate the authors' substantial efforts to improve this manuscript. Owing to their newly generated results from the extensive analyses. I consider this manuscript should have been very much improved. Even though I still have a remaining concern regarding the shallow depth of the WGS presented here and the technical limits of their scCUTseq, I understand that this part should wait for their future work. I sincerely hope the authors further continue the spatial analysis of the cancer cell diversities and will have a change to look back, again, on the results appearing in this particular manuscript.

We thank the Reviewer for appreciating our efforts to improve our manuscript and for their constructive comments and suggestions. We understand the Reviewer's concern about the shallow depth of the WGS data presented and we now explicitly address this issue in the corresponding part in the Results section, in the following newly added sentence:

<< *We note, however, that smaller deletions present in only a few cells might have not been detected in this validation experiment because of the very low sequencing depth achieved.*
>>

We would also like to thank the Reviewer for encouraging us to continue working on the spatial characterization of SCNAs and mutations in cancer: indeed, this is what we are currently doing and we hope to present new insights coming from these continued efforts in the years ahead.